# The loci of environmental adaptation in a model eukaryote

Piaopiao Chen [1,2] & Jianzhi Zhang [1] ✉

While the underlying genetic changes have been uncovered in some cases of adaptive evolution, the lack of a systematic study prevents a general understanding of the genomic basis of adaptation. For example, it is unclear whether protein-coding or noncoding mutations are more important to adaptive evolution and whether adaptations to different environments are brought by genetic changes distributed in diverse genes and biological processes or concentrated in a core set. We here perform laboratory evolution of 3360 *Saccharomyces cerevisiae* populations in 252 environments of varying levels of stress. We find the yeast adaptations to be primarily fueled by large-effect coding mutations overrepresented in a relatively small gene set, despite prevalent antagonistic pleiotropy across environments. Populations generally adapt faster in more stressful environments, partly because of greater benefits of the same mutations in more stressful environments. These and other findings from this model eukaryote help unravel the genomic principles of environmental adaptation.

The past two decades have seen major advances in identifying the likely genetic changes underlying adaptive evolution, including in some of the most iconic cases[1]. While valuable[1–5], for three reasons, these case studies are insufficient for a general understanding of the genomic rules and mechanisms of adaptation. First, because the number of case studies is relatively small, the outcome is dependent on the particular adaptation investigated so may be unrepresentative. Moreover, genetic dissections of adaptations brought by few, large-effect mutations are more likely to be successful than those brought by many, small-effect mutations, potentially distorting our understanding. Second, while each case of adaptive evolution usually involves changes of many phenotypic traits at multiple levels of biological organization, most studies focused on only one trait that is usually not the fitness per se, which may result in a biased view. Third, because our knowledge of the context (*i.e.*, the genetic background and environment) of any past adaptive genetic change is at best incomplete, both the fitness impact of the genetic change and its mechanism of action are typically difficult to reconstruct with certainty. As such, several fundamental questions about the genomic basis of adaptation are unanswered. For example, it has been a long-standing debate whether protein-coding or noncoding mutations are

more important to adaptive evolution[2,5,6]. It is also unclear whether adaptations to diverse environments are brought by genetic changes distributed in diverse genes and biological processes or concentrated in a core set of genes and processes. For instance, estimating the nonsynonymous to synonymous nucleotide substitution rate ratio ($\omega$) in comparative genomics tends to detect positive selection in genes for immunity or reproduction[7], but this enrichment may be attributable to a higher power of the approach in detecting diversifying selection than directional selection instead of an oversized contribution of these genes to adaptation, because diversifying selection tends to result in multiple nonsynonymous substitutions within a codon, facilitating the detection of positive selection.

To understand the genomic rules and mechanisms of adaptation, we adopt the approach of experimental evolution. Unlike the traditional comparative approach that relies on comparing the genotypes and phenotypes of extant organisms adapted to different natural environments, the experimental evolution approach documents genotypic and phenotypic changes of organisms—typically microbes—as they evolve in controlled laboratory environments[8–12]. Hence, both the genetic background and environment of the evolving population are known and the causality between a mutation and an adaptation can

[1]Department of Ecology and Evolutionary Biology, University of Michigan, Ann Arbor, Michigan 48109, USA. [2]Present address: College of Life Sciences, Zhejiang University, Hangzhou 310058, China. ✉e-mail: jianzhi@umich.edu

be unequivocally established. Furthermore, one could identify and validate mutations contributing to an adaptation (*i.e.*, a fitness increase) without being restricted to specific phenotypic traits, thus avoiding potential biases. Because experimental evolution usually lasts for hundreds of generations, while a complete adaptation to an environment could take tens of thousands of generations or longer[13], experimental evolution investigates the initial stage of an adaptation, which is characterized by the most rapid rise in fitness[13,14] realized by fixation of mutations with relatively large benefits[15,16]. Thus, such studies focus on comparatively important genomic changes to adaptation.

In this work, we evolved the budding yeast *Saccharomyces cerevisiae* in a large number of laboratory environments, measured their fitness before and after evolution, sequenced their genomes, and identified and validated putatively adaptive mutations accumulated. These data allow painting a broad picture of the genomic and mechanistic underpinnings of environmental adaptation of a model eukaryote.

## Results

### Experimental evolution
Starting from the same diploid yeast strain, we conducted 800 generations of experimental evolution of 3024 populations by batch culture in 252 different laboratory environments, each with 12 replicates (Supplementary Fig. 1a). The 252 environments (Supplementary Data 1) are divided into two classes (Supplementary Fig. 1b). The first class of 94 environments represent variations in multiple aspects of yeast's natural habitats including commonly found carbon and nitrogen sources, plant and microbial toxins and metabolites, and shifting availability of vitamins and minerals[17]. The second class of 158 environments are chosen from 3250 media containing different drug-like small molecules used in a chemogenomic growth profiling of ~5900 gene deletion yeast strains[18], by maximizing the diversity of environments in our experimental evolution (Supplementary Fig. 1c). To ensure that the yeast adaptations studied here are related to the specific ingredients added or depleted in the media, rather than the common components of the media, we pre-adapted the progenitor strain to the synthetic complete (SC) medium, the base medium in the vast majority of our 252 environments, for 600 generations.

### Fitness changes
The maximum growth rate of yeast under an environment is a major component of its fitness in our experimental evolution under that environment so is used as a proxy for fitness and referred to as fitness unless otherwise noted. Given that we evolved the same progenitor across all environments, the fitness of the progenitor under an environment reflects the degree of environmental challenge to the strain so can be used to measure the environmental stress level, with a lower fitness indicating a higher level of environmental stress. We measured the fitness of the progenitor in each environment used ($F_i$, where $i$ stands for environment $i$) relative to that in SC ($F_{SC}$). Except for 19 environments where precipitates prohibited accurate fitness quantification, $F_i/F_{SC}$ ranged from 0.68 to 1.06, with a mean of 0.88 (Fig. 1a; see Supplementary Fig. 2a where the two classes of environments are separately marked).

To assess the extent of adaptation through experimental evolution in each environment, we measured the mean fitness ($f_i$) of five randomly picked end populations that had evolved in the environment. We found $f_i/F_i$ to exceed 1 in each environment (nominal $P < 0.05$ in 223 of the 233 environments with measurements), with a mean of 1.11 and SD of 0.14 (Fig. 1b). The two classes of environments do not show a significant difference in $f_i/F_i$ (Supplementary Fig. 2b). A strong, negative correlation exists between the relative progenitor fitness in an environment ($F_i/F_{SC}$) and the extent of adaptation ($f_i/F_i$) in the

environment (Pearson's $r = -0.72$, $P = 8.4 \times 10^{-39}$; Fig. 1c; see also Supplementary Fig. 3 for a statistically robust analysis), suggesting that more stressful environments promoted faster adaptation. In theory, this trend could be caused by an increase in the beneficial mutation rate and/or size with the level of environmental stress (see below).

To directly compare the fitness of the end populations from different environments, we calculated $f_i/F_{SC}$, which shows a significant, positive correlation with $F_i/F_{SC}$ (Fig. 1d), suggesting that post-adaptation fitness still tends to be lower in more stressful environments. Notwithstanding, when $F_i/F_{SC}$ and $f_i/F_{SC}$ are both significantly different between two environments (at nominal $P = 0.05$), $f_i/F_{SC}$ is higher in the environment with the relatively low $F_i/F_{SC}$ 14.4% of the time (Fig. 1d). Thus, after adaptation for a given number of generations, fitness sometimes become higher in a more stressful environment than in a less stressful one.

### Genomic changes
To probe the genomic changes during the yeast adaptation, we sequenced the genomes of the progenitor and a single clone from each end population. Hereinafter, we use "substitutions" to refer to the genomic changes identified from the end populations relative to the progenitor, not because these changes were necessarily fixed in the end populations, but because they were subjected to drift and/or selection, unlike "mutations" that refer to genomic changes before the actions of these evolutionary forces. Furthermore, we use "coding" to refer to protein-coding and "genes" to refer to protein-coding genes unless otherwise noted.

The mean number of substitutions per population varied by 58 times across the 252 environments, with a median of 7 (Fig. 2a). The 4-NQO environment saw the highest mean number of 216 substitutions. Three lines of evidence suggest that the rapid genomic evolution under 4-NQO is primarily due to elevated mutagenesis rather than an elevated selective pressure. First, 4-NQO is a highly carcinogenic chemical that induces mutagenesis[19]. Second, $F_i/F_{SC}$ in 4-NQO (0.895) is not particularly low when compared with that across all environments (90% range: 0.733–0.979). Third, the fitness improvement in the adaptation to 4-NQO (8.9%) is not particularly high when compared with that across all environments (90% range: 1.5–42.5%). Observations from several other environments are similar to those in 4-NQO and are likely caused by elevated mutagenesis as well[20–25] (Supplementary Data 2).

In total, we observed 27,056 substitutions in the evolved lines (Supplementary Data 3), allowing a statistical comparison between the relative numbers of various types of substitutions and their neutral expectations (Fig. 2b). For example, the number of substitutions in coding regions relative to that in noncoding regions is 2.90, significantly exceeding the neutral expectation of 2.68 (see Methods). Because negative selection is generally stronger on coding than noncoding regions[26], positive selection must have also been overall stronger on coding than noncoding regions to have yielded a higher substitution rate at coding than noncoding sites in the yeast adaptation.

Consistently, the relative fitness increase in an environment ($f_i/F_i-1$) is more strongly correlated with the number of coding substitutions observed in the environment (Spearman's $\rho = 0.29$, $P = 7.1 \times 10^{-6}$; Fig. 2c) than with the number of noncoding substitutions ($\rho = 0.14$, $P = 0.034$; Fig. 2c). Because the mutation rate varies among environments[27], numbers of coding and noncoding substitutions are strongly positively correlated across the 252 environments ($r = 0.98$, $P = 9.0 \times 10^{-171}$). Nonetheless, the partial correlation between the fitness increase and the number of coding substitutions, after the control of the number of noncoding substitutions, remains significantly positive ($\rho = 0.26$, $P = 6.9 \times 10^{-5}$; Fig. 2c). By contrast, the partial correlation between the fitness increase and the number of noncoding substitutions after the control of the number of coding substitutions is not significant ($\rho = -0.03$, $P = 0.64$; Fig. 2c). These

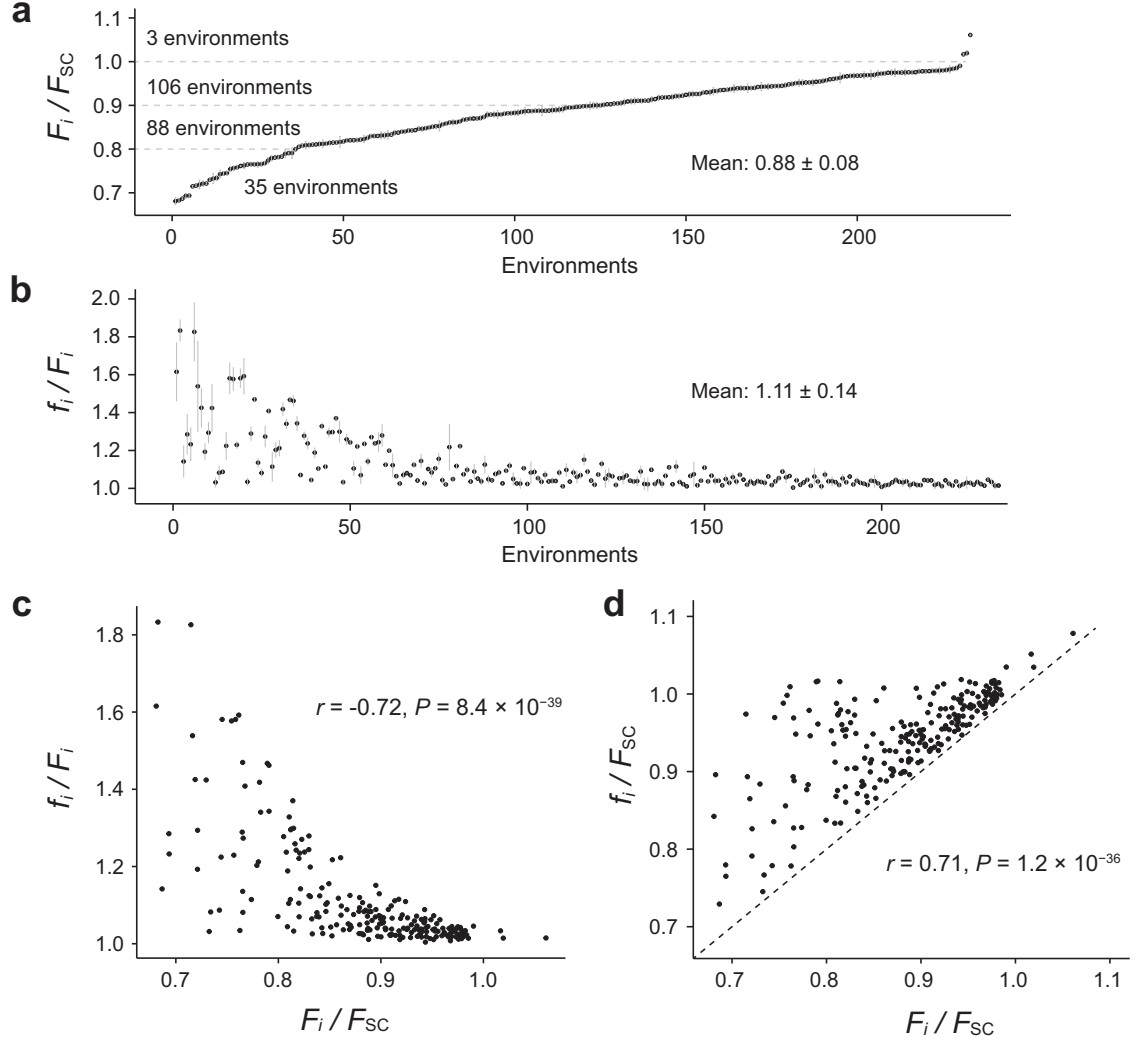

**Fig. 1 | Fitness changes in yeast experimental evolution. a** Fitness of the progenitor in environment $i$ ($F_i$) relative to that in the SC medium ($F_{SC}$) for the 233 environments with reliable measurements. Each dot represents the mean fitness in an environment and the error bar shows the standard error. The mean and standard deviation of $F_i/F_{SC}$ across the 233 environments are also shown. **b** Mean fitness of the end populations after adaptation to environment $i$ ($f_i$) relative to $F_i$. Five evolved populations per environment were measured, with each dot representing the mean and the error bar representing the standard deviation. The environments are ordered as in (**a**). **c** Correlation between the relative progenitor fitness in an environment ($F_i/F_{SC}$) and the extent of adaptation ($f_i/F_i$) in the environment. Pearson's correlation coefficient $r$ and its $P$ value are shown. **d** Correlation between $f_i/F_{SC}$ and $F_i/F_{SC}$. Pearson's correlation coefficient $r$ and its $P$ value are shown. The diagonal line represents no change in fitness in the experimental evolution. Source data are provided as a Source Data file.

results support a major contribution of coding but not noncoding substitutions to the yeast adaptation.

Within coding regions, nonsynonymous mutations are generally subject to stronger negative selection than are synonymous mutations[26]. However, we found the ratio of the number of nonsynonymous substitutions to that of synonymous substitutions to exceed its neutral expectation significantly (Fig. 2b; Supplementary Fig. 4), suggesting an overall stronger positive selection on nonsynonymous than synonymous mutations in our data. Similarly, there is evidence for an overall stronger positive selection on nonsense than synonymous mutations in our data (Fig. 2b; Supplementary Fig. 4).

Frame-shifting insertion/deletion (indel) mutations in coding regions are expected to be under much stronger negative selection than are frame-conserving indels. Yet, the ratio of the number of frame-shifting indels to that of frame-conserving indels significantly exceeds its neutral expectation (Fig. 2b), suggesting an overall stronger positive selection on frame-shifting than frame-conserving indel mutations in the yeast adaptation.

**Putatively adaptive genes, noncoding regions, and substitutions**

Because substitutions could occur through selection, hitchhiking, or drift, we developed a statistical test to identify from each environment and replicate population putatively adaptive genes, which harbor putatively adaptive substitutions (see Methods). Nonsynonymous, nonsense, and frame-shifting indel substitutions are drastically over-represented in the putatively adaptive genes relative to the other genes (Fig. 2b), suggesting that the true adaptive genes and adaptive coding substitutions are highly enriched in the identified sets. Of the 2778 putatively adaptive coding substitutions identified, 97% are nonsynonymous, nonsense, or frame-shifting indel substitutions, whereas only 3% are synonymous or frame-conserving indel substitutions (Fig. 2d).

We subsequently followed the same method to identify putatively adaptive noncoding regions and substitutions (Supplementary Data 4; see Methods). A total of 17 such regions were identified, consisting of eight noncoding regions associated with genes (*i.e.*, untranslated regions and introns if present), seven autonomously replicating sequences (ARSs) in the nuclear genome, one noncoding RNA, and one intergenic region (Fig. 2e). Each of these regions was identified as

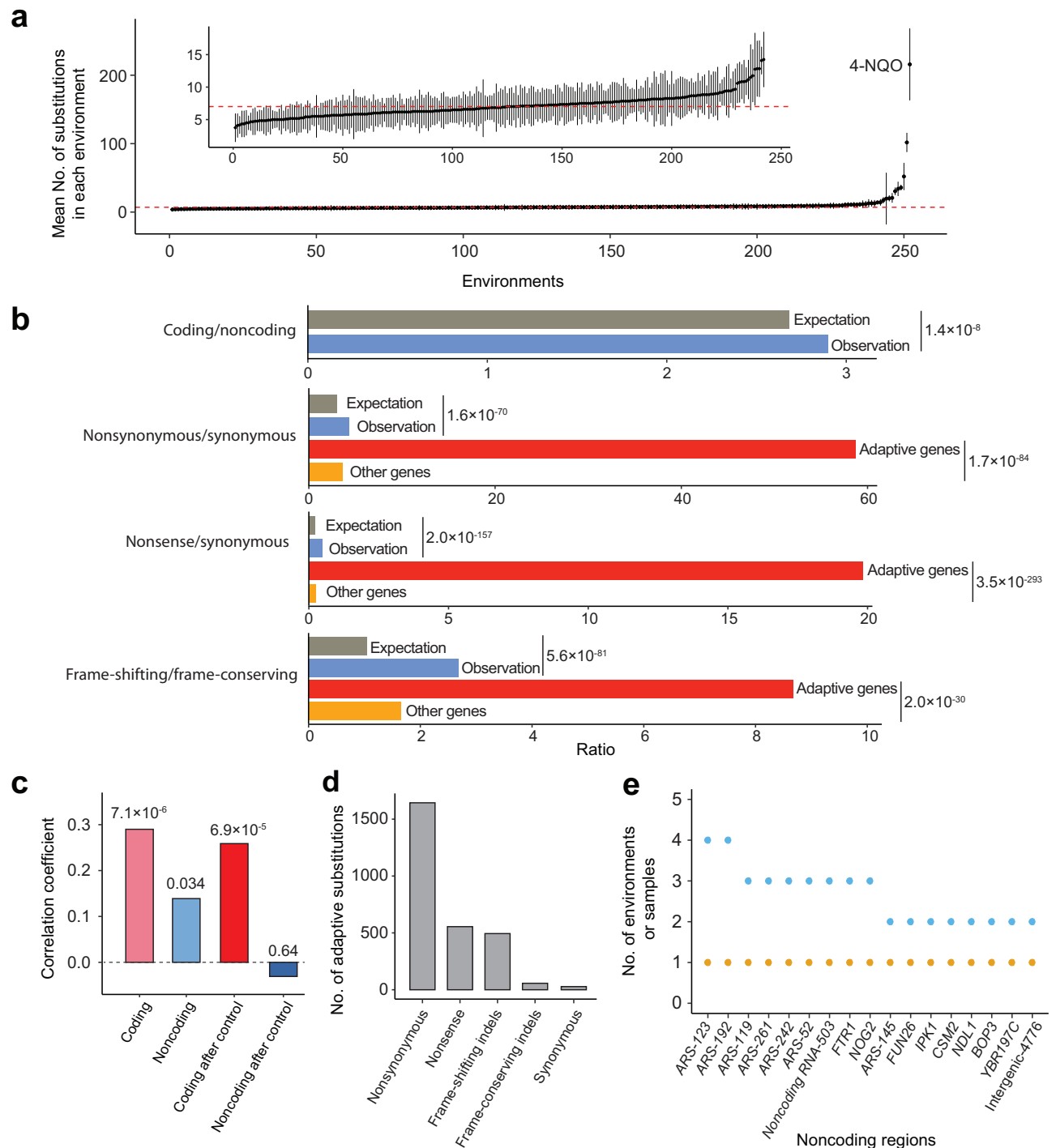

**Fig. 2 | Genomic changes in yeast environmental adaptation. a** Mean number of substitutions across the 12 replicates in each environment. Error bars show standard deviations. The red dashed line indicates the median number of substitutions across the 252 environments. The inset shows the same distribution after the exclusion of ten environments with the most substitutions. **b** Comparison of relative numbers of various types of substitutions. Expectation refers to the neutral expectation. *P*-values are from two-tailed chi-squared tests. **c** Spearman's correlation between the extent of adaptation ($f_i/F_i$) in an environment and the number of coding or noncoding substitutions in the environment (light colors), as well as partial Spearman's correlation between $f_i/F_i$ and the number of coding or noncoding substitutions after the control of the number of noncoding or coding substitutions (dark colors). *P*-value is indicated above each bar. **d** Total numbers of various types of putatively adaptive coding substitutions across all environments. **e** Putatively adaptive noncoding regions. Yellow and blue dots respectively indicate the number of environments and number of samples where the noncoding region is putatively adaptive. Each gene name refers to the untranslated regions and introns (if present) of the gene. ARS, autonomously replicating sequence. Source data are provided as a Source Data file.

adaptive in only one environment (Fig. 2e). Notably, six of the seven adaptive ARSs were identified from the 4-NQO environment, suggesting the possibility that ARS mutations mitigate the inhibition of DNA replication by 4-NQO[28].

Strikingly, the ratio of the number of putatively adaptive coding substitutions (2778) to that of noncoding substitutions (47) is 59.1, substantially greater than the ratio (2.68) of the number of coding sites to that of noncoding sites in the yeast genome ($P = 7.6 \times 10^{-204}$, chi-

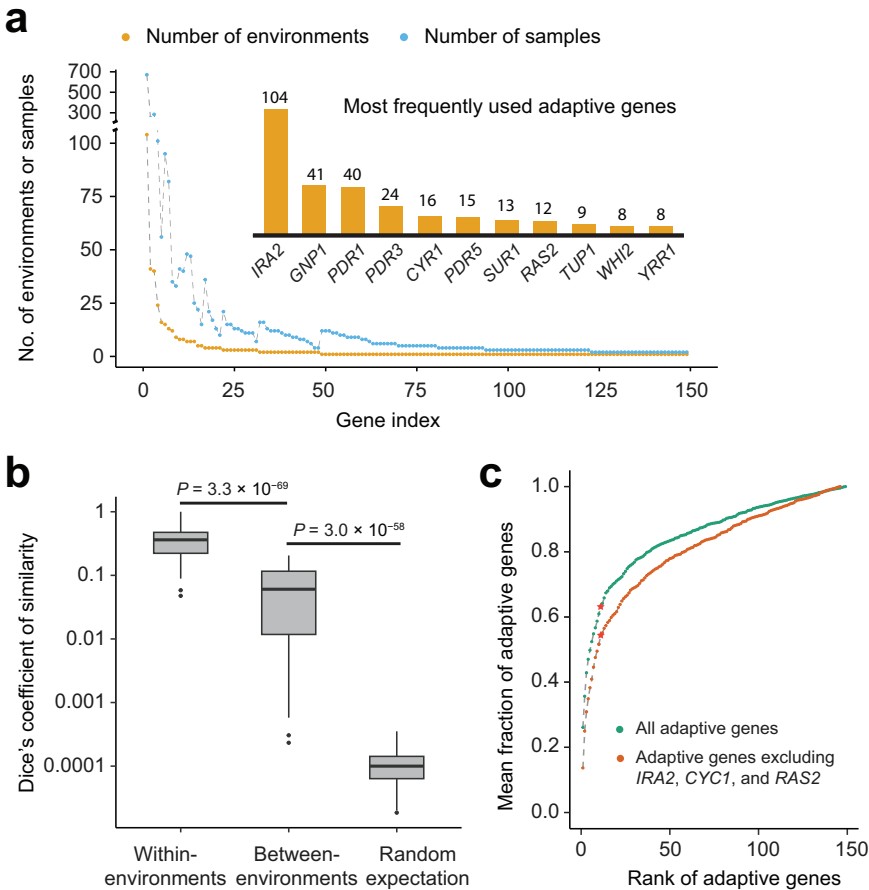

**Fig. 3 | Putatively adaptive genes. a** Numbers of environments or samples in which a gene is identified as adaptive. The 149 adaptive genes are ordered by the number of environments where they are identified as adaptive, with the top 11 genes presented in the inset along with the numbers of environments where they are identified as adaptive. **b** Dice's coefficient of similarity in the identified adaptive genes between samples. The lower and upper edges of a box represent the first (qu₁) and third (qu₃) quartiles, respectively, the horizontal line inside the box indicates the median (md), and the whiskers extend to the most extreme values inside inner fences, md ± 1.5(qu₃-qu₁). Dots show data points outside the whiskers. *P*-values are from two-tailed Wilcoxon rank-sum tests. **c** Cumulative mean fraction of adaptive genes in the 252 environments that are explained by the adaptive genes on the X-axis. Genes are ranked in descending order by their frequency of identification as adaptive genes across all environments. Stars indicate the top 11 adaptive genes. Source data are provided as a Source Data file.

squared test), supporting the primary contribution of coding substitutions to the yeast adaptation.

### Sharing of adaptive genes among environments

Because our yeast adaptation was overwhelmingly fueled by coding mutations, the rest of the study focuses on adaptive genes. We identified between 0 and 6 adaptive genes per environment, with a median of 2, across the 252 environments (Supplementary Fig. 5a). In total, 149 distinct adaptive genes were detected; 32% of them were adaptive in multiple environments, with *IRA2*, *GNP1*, and *PDR1* being adaptive in 104, 41, and 40 environments, respectively (Fig. 3a). Clearly, the propensity for being adaptive varies substantially among genes (Supplementary Fig. 5b).

We employed Dice's coefficient of similarity to measure the fraction of adaptive genes shared between two evolved lines (see Methods). Even in adaptations to different environments, the sharing of adaptive genes is 600 times the random expectation; adaptations to the same environment further raise the sharing of adaptive genes by 6.3 times (Fig. 3b). No significant difference in mean Dice's coefficient was found between different environments of the same group and those of different groups described in Supplementary Fig. 1b (Supplementary Fig. 5c). By increasing the number of replicate populations in experimental evolution from 12 to 96 in four environments (Supplementary Fig. 5d), we confirmed that the substantial sharing of

adaptive genes between environments is robust to the number of replicates (Supplementary Fig. 5e). As expected, with an increased detection power bestowed by 96 replicates, we observed more adaptive genes (8 to 15 per environment, with a median of 11.5 across the four environments).

We estimated the overall contribution of each identified adaptive gene to the adaptations in the 252 environments (see Methods) and found this value extremely unequal across genes. The top 11 most frequently used adaptive genes, constituting only 7.4% of all adaptive genes identified (Fig. 3a), account for an average of 63% of adaptive genes per environment (Fig. 3c). To assess the impact of the potential further adaptation of the progenitor to the common components in the diverse media used, we evolved the progenitor in SC for another 800 generations, with 96 replicates, which led to the identification of three adaptive genes (*IRA2*, *CYC1*, and *RAS2*). Even after the exclusion of these three genes, the top 11 most frequently used adaptive genes in the 252 environments account for an average of 54% of adaptive genes per environment (Fig. 3c). Taken together, our results indicate that a small set of genes are repeatedly used for adaptation to diverse environments.

### Features of adaptive genes

Based on the fitness defects of gene deletions in SC[29], we found adaptive genes to be functionally more important than other genes

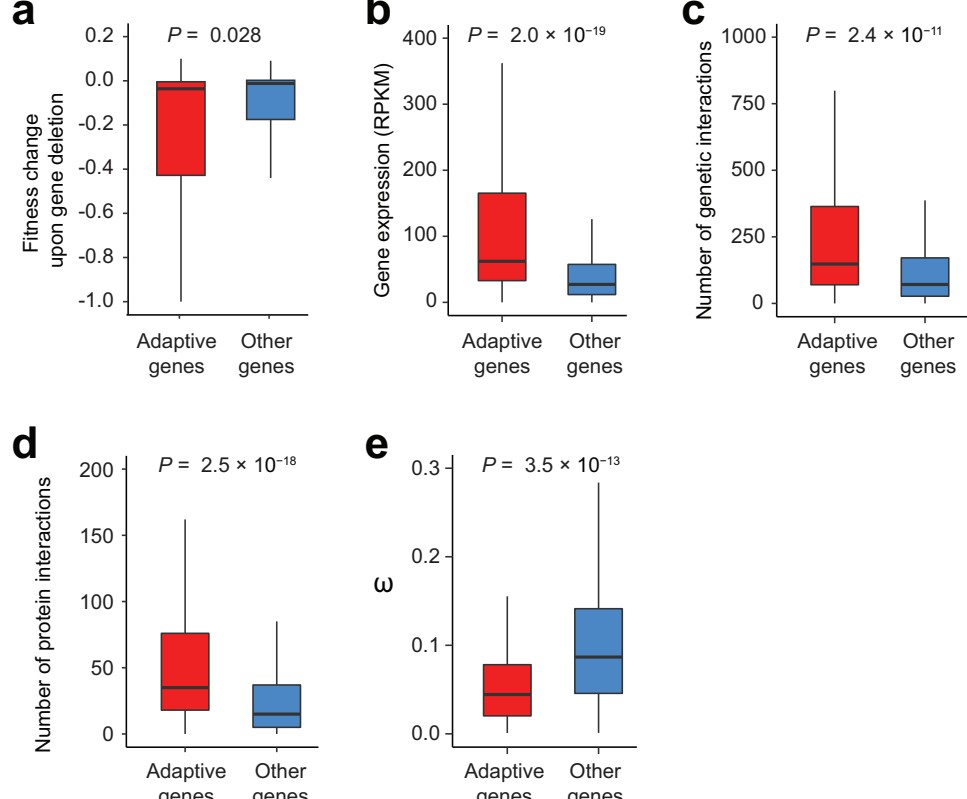

**Fig. 4 | Properties of putatively adaptive genes.** Compared with other genes, adaptive genes cause larger fitness drops upon deletion (**a**), are more highly expressed (**b**), engage in more genetic (**c**) and protein-protein (**d**) interactions, and have lower $\omega$ between *S. cerevisiae* and *S. paradoxus* (**e**). The lower and upper edges of a box represent the first (qu$_1$) and third (qu$_3$) quartiles, respectively, the horizontal line inside the box indicates the median (md), and the whiskers extend to the most extreme values inside inner fences, md $\pm 1.5$(qu$_3$-qu$_1$). *P*-values are from two-tailed Wilcoxon rank-sum tests. In (**b**), RPKM, Reads Per Kilobase per Million mapped reads. Source data are provided as a Source Data file.

(Fig. 4a). Adaptive genes tend to have higher expression levels than other genes (Fig. 4b). They also possess a significantly higher number of genetic interaction partners (Fig. 4c) and protein interaction partners (Fig. 4d) than other genes. As mentioned, in interspecific comparisons, $\omega$ estimated for a gene is frequently used for detecting selection acting on the gene[26]. Positive and negative selections are respectively inferred when $\omega$ is significantly higher and lower than 1. While it is known that $\omega$ computed from the entire coding sequence of a gene often lacks power for detecting positive selection due to the occurrences of both positive and negative selections, the relationship between $\omega$ and the gene's propensity for contribution to adaptation is unclear. Interestingly, $\omega$ computed from the orthologous gene sequences of *S. cerevisiae* and its sister species *S. paradoxus* tends to be lower for our identified adaptive genes than the other genes in the yeast genome (Fig. 4e).

We performed a Gene Ontology (GO) analysis of the 149 identified adaptive genes. In terms of biological processes, 28 of these genes participate in signal transduction, four times the random expectation ($P = 7 \times 10^{-7}$, hypergeometric test adjusted for multiple testing by the Bonferroni correction; Fig. 5a). These 28 genes harbor 37% (1,026/2,778) of all putatively adaptive substitutions, and the majority (83%) of these 1,026 substitutions are in nine genes involved in Ras protein signal transduction (Fig. 5b). In addition, 38 putatively adaptive genes participate in transmembrane transport or chemical homeostasis (genes overlap substantially in these two GO terms), three to five times the random expectation (Fig. 5a, b). The identified adaptive genes are also significantly enriched in several other biological processes (Fig. 5a; Supplementary Fig. 6). Consistent with these enriched biological processes are the enrichment of transporter activity, transcription

regulator activity, and GTPase activity among molecular functions and the enrichment of membrane and cell periphery among cellular components (Fig. 5a).

Considering the biological processes affected by the identified adaptive genes, we found that alteration in signal transduction is involved in 136 of the 252 environmental adaptations, followed by alterations in positive regulation of transcription (98), transmembrane transport (82), chemical homeostasis (75), and others (Fig. 5c). Clearly, a few biological processes play predominant roles in diverse environmental adaptations.

Finally, we compared the adaptive genes identified in the two large classes of environments used in our experimental evolution (Supplementary Fig. 1). We observed a higher prevalence of PDR genes (*PDR1*, *PDR3*, and *PDR5*) in Class II environments, whereas signal transduction genes such as *IRA2*, *CYR1*, and *RAS2* are enriched in Class I environments. Specifically, 34% (52) of Class II environments but only 3% (3) of Class I environments used PDR genes ($P = 1.1 \times 10^{-7}$, chi-squared test). Conversely, 66% of Class I environments but only 52% of Class II environments used signal transduction genes ($P = 0.065$). Further subdivision of Class I environments into five groups (Supplementary Fig. 1b) revealed a propensity for transmembrane genes in environments with different nitrogen resources, notably *GNP1*. Specifically, 41% (9) of environments with different nitrogen resources utilized *GNP1*, compared to only 15% in all other environments studied ($P = 0.0052$).

### Fitness effects of adaptive substitutions
To verify that the identified adaptive genes and substitutions indeed contributed to the relevant yeast adaptation, we used CRISPR/Cas9 genome editing to introduce a subset of substitutions in the progenitor.

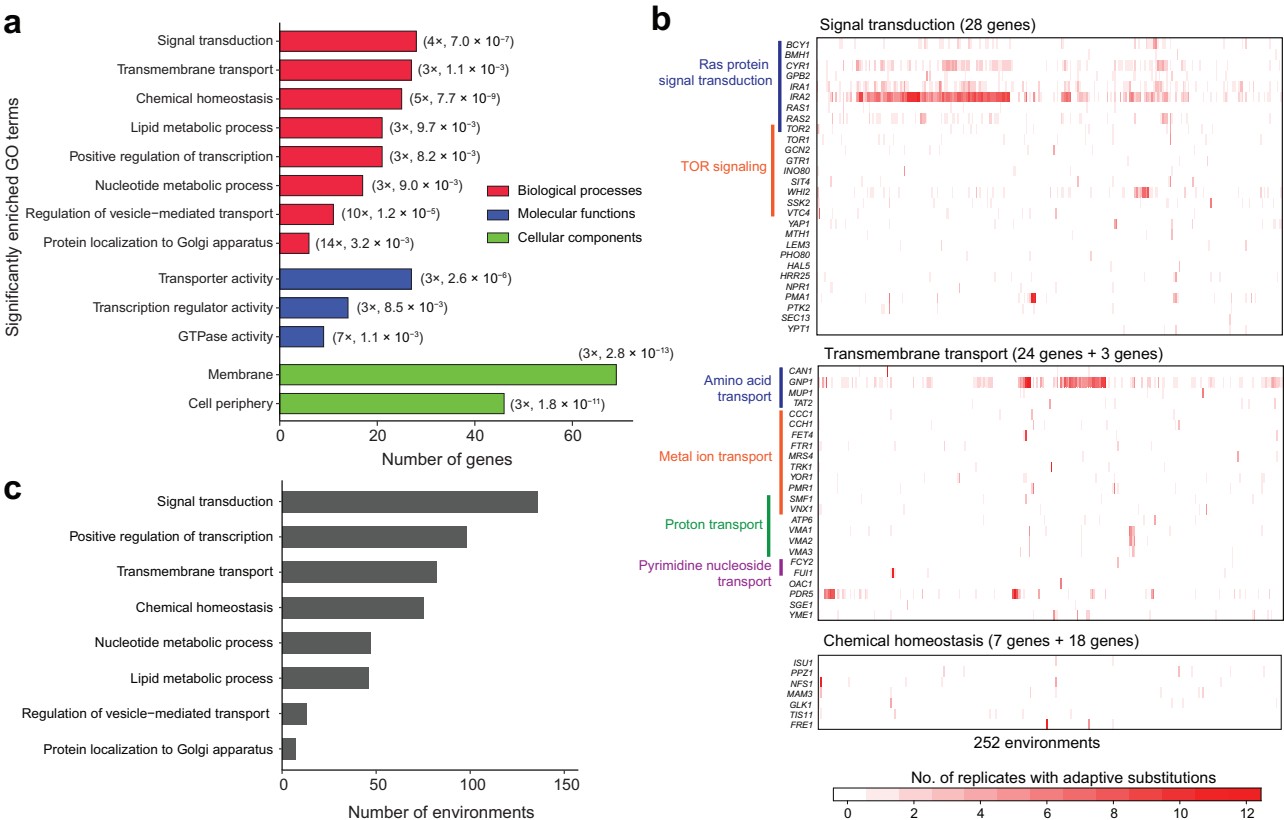

**Fig. 5 | Biological processes frequently involved in environmental adaptation.** **a** Gene Ontology (GO) terms significantly enriched in the 149 putatively adaptive genes. In the parentheses are the fold enrichment followed by Bonferroni corrected *P*-value. **b** The top three enriched biological processes in putatively adaptive genes. Each column in the heatmap represents an environment and each row shows one adaptive gene. Environments are ordered based on a clustering analysis of all putatively adaptive genes. Color indicates the number of replicates in the environment that harbor adaptive substitutions in the gene. In the parentheses are the number of genes shown + the number of genes shared between the focal GO term and other GO terms and are shown under other GO terms. More specific GO terms enriched in adaptive genes are also listed alongside. **c** Most frequently altered biological processes in environmental adaptation and the numbers of environments where they are altered. Source data are provided as a Source Data file.

For each gene of interest, we randomly chose ten "home" environments, where the gene was identified as adaptive, and ten "foreign" environments, where no substitution was observed in the gene. Note that different genes typically have different home/foreign environments. We then quantified the fitness effects of each substitution across these 20 environments (Supplementary Data 5). This analysis could help understand why a gene was adaptive in some but not other environments.

*IRA2* is a putatively adaptive gene in 104 environments. Substitutions in *IRA2* are distributed throughout its coding region, primarily consisting of nonsense (267, 38%) and frame-shifting indel (266, 38%) substitutions (Fig. 6a), suggesting that a loss of *IRA2* function is advantageous in these 104 environments. We created a one-nucleotide deletion in the coding region of *IRA2* in the progenitor and found that this deletion significantly increased yeast fitness in each of the 10 home environments examined (Fig. 6b). By contrast, in eight of the ten foreign environments investigated, the deletion significantly reduced yeast fitness, while in the remaining two foreign environments, the deletion had no significant fitness effects (Fig. 6b). These results confirm that *IRA2* was adaptive in the home environments. Furthermore, because the loss of *IRA2* function is beneficial in many but deleterious in many other environments, *IRA2* cannot be universally adaptive despite that it is the most common adaptive gene found.

*PDR1* was identified as an adaptive gene in 40 environments. Substitutions in *PDR1* are enriched in certain coding regions, with 279 of 288 substitutions being nonsynonymous and only eight being nonsense or frame-shifting (observed in three environments) (Fig. 6c). This substitution pattern, contrasting that in *IRA2*, suggests that a loss

of *PDR1* function is beneficial in only a few environments whereas a gain of function is beneficial in many environments. We introduced a frequently observed *PDR1* nonsynonymous substitution in the progenitor. This substitution increased the fitness significantly in eight of the ten home environments examined, increased the fitness nonsignificantly in the ninth environment, and decreased the fitness nonsignificantly in the tenth environment (Fig. 6d). Notably, this tenth environment is one of the three environments where loss-of-function substitutions occurred in *PDR1*. Hence, it is plausible that, even in the same gene, depending on the environment, different types of substitutions and functional changes contributed to adaptation. By contrast, in six of the ten foreign environments, the substitution significantly lowered fitness (Fig. 6d).

Our results from *IRA2* and *PDR1* indicate the presence of antagonistic (environmental) pleiotropy[30] of both loss-of-function and gain-of-function substitutions observed in the identified adaptive genes. To test the generality of this finding, we investigated one substitution in each of six additional frequently used adaptive genes. The results (Supplementary Fig. 7a–l) are similar to those from *IRA2* and *PDR1*. Specifically, the sign of the fitness effect of the substitution is more often positive in the home than foreign environments for all eight genes studied, and this disparity is statistically significant for seven genes (Fig. 6e). When the results from the eight substitutions are combined, the substitutions show positive fitness effects (regardless of the significance of the effect) in 66 of 76 home environments but negative effects in 62 of 80 foreign environments (Fig. 6e).

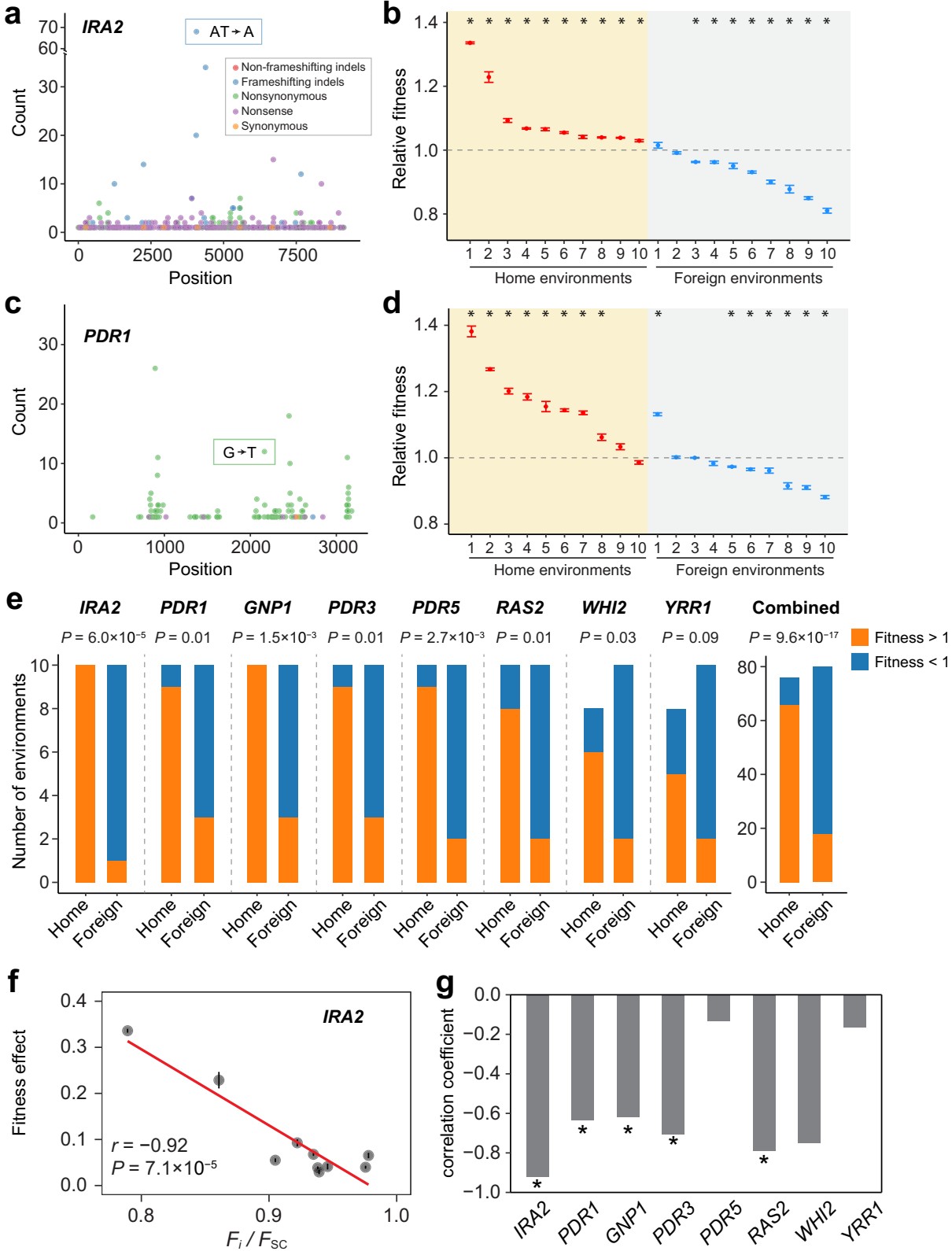

A similar trend is seen when only significant fitness effects are considered (Supplementary Fig. 7m).

**Larger effects of the same beneficial mutations in more stressful environments**

What might have caused the generally faster yeast adaptation in more stressful environments (Fig. 1c)? First, yeast mutation rate per generation increases with the level of environmental stress[27], predicting a higher beneficial mutation rate in more stressful environments under the reasonable assumption that the fraction of mutations that are beneficial does not decrease with the level of environmental stress. Second, it has been suggested that the effect size of a beneficial mutation increases with the environmental stress level[31], which we can test using the eight mutations studied in the preceding section.

**Fig. 6 | Fitness effects of putatively adaptive substitutions. a** Adaptive substitutions along *IRA2*. Count refers to the number of adaptive substitutions observed across all samples. A one-nucleotide deletion (boxed) in the coding region of *IRA2* was introduced into the progenitor strain. **b** Fitness of the progenitor with the deletion relative to that without the deletion in ten home and ten foreign environments. **c** Distribution of adaptive substitutions along *PDR1*. A G-to-T nonsynonymous substitution (boxed) in *PDR1* was introduced into the progenitor. **d** Fitness of the progenitor with the nonsynonymous substitution relative to that without the substitution in home and foreign environments. In (**b**, **d**), each dot represents the mean relative fitness from three replicates, with error bars indicating the standard error; *P*-values are from one-sample two-tailed *t*-tests and are indicated by * (*P* < 0.05). **e** Numbers of home and foreign environments where the fitness of the mutated progenitor relative to the fitness of the progenitor is > 1 or < 1 for each mutant constructed. *P*-values are from one-tailed Fisher's exact tests. **f** Larger benefits of the one-nucleotide deletion in *IRA2* under more stressful environments. Each dot represents the mean fitness effect in a home environment, with error bars indicating the standard error. The line shows the linear regression, with Pearson's correlation (*r*) and one-tailed *P*-value presented. $F_i$ used for computing the fitness effect and in $F_i/F_{SC}$ are separately measured. **g** Correlation (*r*) between the fitness effect and $F_i/F_{SC}$ across the examined home environments where the substitution is not significantly deleterious, for each of the eight substitutions. *, one-tailed *P* < 0.05. Source data are provided as a Source Data file.

Indeed, the benefit of the one-nucleotide deletion in *IRA2* rises with the stress level of the home environment (Fig. 6f). Strikingly, this trend is observed for all eight mutations, a highly nonrandom pattern (*P* = 0.0078, two-tailed binomial test; Fig. 6g; Supplementary Fig. 8).

## Discussion

We found that population fitness rises faster in more stressful environments and that this phenomenon is explained by both an increase in the beneficial mutation rate[27] and an increase in the beneficial mutation size with the environmental stress level. The latter trend is analogous to the widely reported trend that the rate of fitness increase of an adapting population declines as the population adapts to its environment, owing to diminishing returns epistasis[13,14,16,31–35]. These previous studies, along with the present research, demonstrate the context-dependency of the fitness effects of mutations, where the previous studies focused on the genetic context while our study focused on the environmental context. Whether these two trends have the same mechanistic basis (*e.g.*, idiosyncrasy in both gene-environment and gene-gene interactions[16,36]) awaits further research.

Our finding of a substantially greater contribution of coding than noncoding substitutions to the yeast adaptation seems at odds with reports of a higher prevalence of adaptive substitutions at noncoding than coding sites from population and comparative genomic studies of flies and mice[37,38]. There could be several explanations for this discrepancy. First, the truth may differ between the unicellular fungus and multicellular animals given that noncoding regions constitute a larger fraction of the genome and potentially play a more important, developmental role in animals than in yeast. Second, the discrepancy could be artifactual because the earlier results relied on simplifying assumptions that are now known to be violated (*e.g.*, neutrality of synonymous mutations[39–42] and demographic constancy[43]). By contrast, our inference is not dependent on these assumptions. Third, our experimental evolution lasted for 800 generations and accumulated a median of seven substitutions, thus representing only early stages of adaptation. The fitness gain from each consecutive substitution is expected to decrease during an adaptation[15,16]. If beneficial coding mutations on average have larger fitness effects than beneficial noncoding mutations[37], our data collected from the first steps of adaptation would enrich coding relative to noncoding substitutions when compared with adaptive substitutions inferred from interspecific comparisons. Notwithstanding, because a large fraction of the fitness gain in adaptation to an environment is realized in early stages (see Methods), at the minimum, our data suggest that, during adaptation, coding substitutions are more important than noncoding substitutions in raising the fitness.

Several characteristics of the adaptive genes discovered here appear to contradict those (*e.g.*, fast evolution) of adaptive genes identified from comparative genomics. One potential explanation is that comparative genomics typically uses the criterion of rapid sequence evolution (*e.g.*, *ω* > 1) to identify adaptive genes, thus missing slowly evolving adaptive genes. Because many gene properties are intercorrelated (*e.g.*, high expression is correlated with slow sequence evolution[44] and more protein interactions[45]), this practice would also bias other characteristics of adaptive genes. Additionally, because mutations in functionally important genes should on average have larger fitness effects, either negatively or positively, than mutations in other genes, our focus on early stages of adaptation tends to identify adaptive genes of relatively high functional importance, as explained earlier.

We found that yeast repeatedly employed a small set of genes and modified a core group of biological processes including signal transduction, transcriptional regulation, transmembrane transport, and chemical homeostasis to cope with diverse environmental challenges. Over 50% of adaptive genes in any environment studied are expected to fall in the 11-gene core set. Some of these adaptive genes such as *IRA2*, *PDR5*, and *WHI2* were also discovered in previous yeast evolution experiments[46–48]. Notably, disrupting *IRA2*, a negative regulator of Ras protein signal transduction, conferred a large fitness advantage and was adaptive in 104 diverse environments. Further, nine of the 57 Ras pathway genes are putatively adaptive in our data and three of them are in the core gene set. This is reminiscent of the recent report that 46% of tumors have mutations in the RTK/RAS pathway, the highest among all oncogenic signaling pathways[49]. *NF1*, the human homolog of *IRA2*[50], harbors loss-of-function mutations in 5% of all tumors[49]. Promoting cell growth by activating the Ras pathway is apparently a commonly used mechanism for both yeast adaptive evolution[47] and human tumorigenesis. This said, we discovered that disrupting *IRA2* is deleterious in many environments where *IRA2* was not found to be adaptive, indicating that disrupting *IRA2* can be beneficial or harmful depending on the environment. Consistent with this observation are the report of multiple independent origins of loss-of-function alleles of *IRA2* in *S. paradoxus*[51] as well as evidence for balancing selection on *IRA2* and other genes in the Ras pathway in both *S. cerevisiae* and *S. paradoxus*[52]. More broadly, by measuring the fitness effects of eight putatively adaptive substitutions in multiple home and foreign environments, we detected widespread antagonistic pleiotropy, echoing observations of common antagonistic pleiotropy of segregating polymorphisms and null mutations in yeast[29,53]. Therefore, despite the frequent use of the core genes in various environmental adaptations, adaptation to each environment must favor certain core genes and avoid other core genes. Antagonistic pleiotropy explains why the adaptive substitutions observed under certain environments in our study are not present in the progenitor strain or fixed in nature[30].

Our study has several limitations. First, although we attempted to maximize the diversity of the environments used, some variables in yeast's natural and potential environments were excluded (*e.g.*, temperature, altitude, sexual reproduction) due to the difficulty in implementing them in a high-throughput experiment. Future studies should examine to what extent our conclusions based on the adaptations to the 252 diverse environments are general, particularly in scenarios that involve sexual reproduction.

Second, our study focused on the first 800 generations of each environmental adaptation, which dictates that the fitness effects of the substitutions we observed are on average larger than those of substitutions that would occur later in the adaptation. For example, 97% of the putatively adaptive coding substitutions we identified are

nonsynonymous, nonsense, and frame-shifting indel substitutions. Compared with synonymous and frame-conserving indel mutations, nonsynonymous, nonsense, and frame-shifting indel mutations cause greater changes to the gene product so likely confer larger fitness effects both when they are deleterious and when they are beneficial. Hence, it is expected that nonsynonymous, nonsense, and frame-shifting indel substitutions are more common in early than late stages of adaptation. Future longer laboratory evolution experiments may help assess to what degree later stages of adaptation alter the genomic patterns and mechanisms unraveled here. In this context, it is worth mentioning that the types of substitutions and their relative abundances reported from other experimental evolution studies[48,54–57], including 10,000 generations of *S. cerevisiae* adaptations to three environments[48] and 60,000 generations of *Escherichia coli* adaptation to one environment[55], are not substantially different from our observations.

Third, virtually no natural environment is constant, yet the experimental evolution in our study occurred in constant environments. In this sense, our experiment is unrealistic. However, studying adaptations to changing environments is more complex and requires first understanding the principles and mechanisms of adaptation to constant environments[58]. Furthermore, if the environmental change is infrequent, adapting to a changing environment may be viewed as consecutive, incomplete adaptations to a series of constant environments, especially given that our data were collected in the early stage of the adaptation to each constant environment. Hence, our data would be valuable for comparisons with data to be collected in future laboratory adaptation to changing environments.

Finally, our experimental evolution investigated the evolution of only one strain. However, we do not believe that this design compromised our conclusions, because the observations (*e.g.*, patterns of fitness and genomic changes) are generally consistent with those from previous microbial experimental evolution studies that together employed many different strains[8–11,46,48,55–57,59]. This said, it is worth noting that some parameters not accounted for in our setup (*e.g.*, population size) may influence adaptation[60] and thus could affect the generality of our findings. Most importantly, it is unclear whether our conclusions from a unicellular eukaryote are applicable to multicellular eukaryotes, which often possess a much larger fraction of the noncoding genome. Nevertheless, given the high number of unicellular species and organisms on Earth and their many fundamental roles in the ecosystem (including impacts on human health and disease), our findings from the budding yeast are of broad relevance.

## Methods

### Strains and media for yeast experimental evolution
In an earlier experiment, we evolved the diploid yeast strain BY4743 in a synthetic complete (SC) medium with 2% glucose for 600 generations, with three replicates. We then randomly picked three colonies from each replicate population and measured their maximum growth rates in the SC medium using BioTek Gen5 Microplate Reader. The strain with the highest maximum growth rate was used as the progenitor in the present experimental evolution.

The 252 media utilized in this study (Supplementary Data 1) belong to two classes. The first class of 94 environments were derived from 200 environments previously used for growth characterizations of natural strains of *S. cerevisiae* and related species[17]. These 200 environments represent variations in multiple aspects of yeast's natural habitats including commonly found carbon and nitrogen sources, plant and microbial toxins and metabolites, and shifting availability of vitamins and minerals[17]. Because some of the 200 environments contain different concentrations of the same additives and therefore are similar to one another, only 94 of them carrying different additives were used in our study. The second class of 158 media were chosen from ref. 18. encompassing diverse drug-like small molecules. In ref. 18.

authors measured yeast fitness of ~1100 heterozygous gene deletion strains and ~4800 homozygous gene deletion strains in response to 3250 compounds. We first picked 558 commercially available compounds. We then correlated the fitness values across all deletion strains for each pair of compounds. Based on the resulting correlation matrix, we conducted a hierarchical clustering with complete linkage using the heatmap.2 function in the "gplots" package. Under the cutoff of height = 4, we identified 10 groups of compounds with relatively high within-group intercorrelations (Supplementary Fig. 1c). A total of 158 environments were then chosen to ensure that every group was included, maximizing the diversity of the media to be used in our experimental evolution. We adjusted chemical concentrations in the media to impose non-lethal stresses on yeast growth while also ensuring that the doubling time in each environment is shorter than three hours. The base medium of all media used was SC or occasionally the YPD medium if ingredients in SC react with the additive and cause precipitation. Because the progenitor had been (partially) preadapted to SC, most subsequent beneficial mutations accumulated in the experimental evolution should be related to the specific ingredients added or depleted in the 252 media rather than the common components of the SC medium. For media with chemicals needed to be stored at −20 °C, we prepared fresh media every two weeks to avoid degradation.

### Experimental evolution
The 3024 (252 environments × 12 replicates) parallel serial transfer experiments were all initiated from the same overnight culture from a clone of the progenitor strain. For each population, we grew 500 μl of yeast culture in an incubating shaker at 220 rpm and 30 °C. We used plates of 96 deep wells to perform the experimental evolution. Every 24 h, after the culture had reached the stationary phase, we transferred 2 μl of stationary culture (around $10^5$ cells) into 500 μl fresh culture medium using Integra VIAFLO 96-channel handheld electronic pipette. We carried out 100 such transfer cycles for a total of 800 generations. At the end of the experiment, both the populations and a single clone picked from each population were stored at −80 °C for future analysis. To assess the impact of the number of experimental evolution replicates on Dice's coefficient of similarity, we performed experimental evolution of 84 additional replicates in each of four environments (low glucose, NaCl, pH 3, and 5% ethanol) chosen from the 252 environments. To exclude the possibility that the genetic changes observed in our experimental evolution reflected adaptation to the base medium rather than the ingredients that differ among the 252 media, we conducted experimental evolution of the (SC-preadapted) progenitor for 800 additional generations in SC, with 96 replicates. In total, we carried out experimental evolution of 3456 populations in 253 environments.

### Fitness assays
Cells from frozen cultures were inoculated in 500 μl SC medium and incubated for 24 h at 30 °C. The cells were then precultured overnight until saturation in the medium to be tested. Cultures were diluted to an optical density (OD) of 0.01–0.02 in 100 μl of fresh medium and cultivated for 24 h in a Costar 96-well plate under continuous shaking at 30 °C using BioTek Gen5 Microplate Reader. The OD was measured every 10 min. OD measurements were converted to cell densities according to a pre-determined OD-cell density curve. Slopes were calculated for every seven measurements spaced 60 min apart along the growth curve by Δln(cell density)/60. The maximum growth rate was calculated following an existing protocol[61]. In brief, the two highest slopes were discarded to exclude potential artifacts, and a mean slope representing the growth rate per min was calculated from the third to eighth highest slopes. We first measured the maximum growth rate of the progenitor in SC, followed by the calculation of its mean maximum growth rate ($\overline{R_{SC}}$) using data from six replicates. To quantify

the level of stress imposed by each environment, we measured the maximum growth rate of the progenitor in each of the 252 environments ($R_i$, where $i$ stands for environment $i$), with three replicates. The fitness of the progenitor in environment $i$ relative to that in SC was estimated by $F_i/F_{SC} = 2^{R_i/R_{SC}-1}$, with its mean and standard error (based on the three $R_i$ measurements) reported. To assess the extent of adaptation, we randomly chose five end populations per environment and measured their maximum growth rates in the relevant environment with one replicate per population ($r_i$). The fitness of an evolved population relative to $F_i$ was estimated by $f_i/F_i = 2^{r_i/\overline{R_i}-1}$, where $\overline{R_i}$ is the mean maximum growth rate of the progenitor in environment $i$. Because the extent of adaptation was measured in five end populations per environment, the reported $f_i/F_i$ represents the mean ± standard deviation from the five populations.

In analyzing the relationship between $f_i/F_i$ and $F_i/F_{SC}$ across environments, to prevent an artificial negative correlation between these variables caused by the measurement error of $F_i$[62,63], we used the first two of three measurements of $F_i$ to compute $F_i/F_{SC}$ and the third measurement of $F_i$ to compute $f_i/F_i$ in Supplementary Fig. 3.

## Library construction and genome sequencing

The progenitor and 3456 clones, one from each of the end populations in the experimental evolution, were genome sequenced. For each strain, after overnight growth in the SC medium, genomic DNA was extracted from around $10^7$ yeast cells using a MasterPure Yeast DNA Purification Kit (Lucigen; MPY80200). Sequencing libraries were constructed using Nextera DNA Flex Library Prep (Illumina; 20018705). Samples were sequenced using an Illumina HiSeq X with a paired-end 150-nucleotide strategy. Approximately 2 million read pairs were generated from each library, corresponding to an average sequencing depth of ~40×.

## Identification of genomic changes

Sequencing reads were aligned to the *S. cerevisiae* reference genome (R64-3-1) using Burrows-Wheeler Aligner[64] with default parameters, and duplicate reads were removed using Picard tools (http://broadinstitute.github.io/picard/). Single nucleotide variants (SNVs) and indels were called using the Genome Analysis Toolkit (GATK) platform[65]. Each variant had to be supported by at least five reads. All identified SNVs and indels are listed in Supplementary Data 3.

## Assessment of cross-contamination

A shared substitution among strains sampled from different evolved populations could be caused by cross-contamination, occur by chance, or result from a common selective pressure. In our experimental evolution, the 12 replicate populations per medium were respectively placed in the 12 wells of the same row of a plate, whereas different rows had different media. Because the probability of cross-contamination should be higher within than between plates, we hypothesize that cross-contamination should lead to more shared substitutions among samples from different rows of the same plate than among samples from different plates. For every pair of genomes of the evolved strains, we calculated their number of shared substitutions relative to the mean number of substitutions in the two genomes. We found that this value ranged from 0 to 0.05 (mean = 0.002735) between rows within plates ($X$) and ranged from 0 to 0.08 (mean = 0.002727) between plates ($Y$), respectively. That no significant difference exists between $X$ and $Y$ ($P = 0.65$, two-tailed Wilcoxon rank-sum test) and that $X$ and $Y$ are both extremely small strongly suggest that potential cross-contaminations in our data are negligible.

## Identification of putatively adaptive genes

Of the 26,870 substitutions identified, 20,119 fell in coding regions. We attempted to identify putatively adaptive substitutions in coding and noncoding regions separately. Because the number of genes in the yeast

genome is about two orders of magnitude greater than the median combined number of substitutions in the 12 replicate populations per environment, without positive selection, it is highly unlikely that the same gene is hit by more than one substitution in the 12 samples of each environment. This is why conventionally all multi-hit genes are treated as putatively adaptive genes and all substitutions in multi-hit genes are treated as putatively adaptive substitutions[11,12]. However, a very long gene could by chance experience more than one hit without the influence of positive selection. Hence, to be rigorous, we computed the probability that a gene harbors more substitutions than the chance expectation in an environment. Specifically, for each environment and each sample, we randomly generated the same number of substitutions in coding regions as the observed number. Considering all randomly generated substitutions in the 12 samples of the environment, we counted the number of substitutions in each gene. Thus, longer genes had proportionately more random substitutions. The simulation was repeated 100,000 times, and the fraction of repeats in which the number of randomly generated substitutions equals to or exceeds the observed number is the $P$-value for the gene. A gene is then regarded as a putatively adaptive gene if its adjusted $P$-value is smaller than 0.05. The adjusted $P$-value is calculated using Bonferroni correction where the number of tests equals the number of genes in the genome. In each environment, substitutions in putatively adaptive genes are referred to as putatively adaptive substitutions in that environment. Note that our method does not guarantee the finding of all adaptive genes and adaptive substitutions in an environment, because an adaptive gene may be used only once in the 12 replicates and thereby escape our detection.

## Identification of putatively adaptive noncoding regions

The method used for identifying putatively adaptive noncoding regions and substitutions is the same as described in the preceding section except that the unit of analysis is a noncoding region instead of the coding region of a gene. Noncoding regions are the entire nuclear genome subtracting protein-coding regions. These regions include 5' and 3' untranslated regions (UTRs), introns, noncoding RNAs, centromeres, telomeres, autonomously replicating sequences (ARSs), long terminal repeats (LTRs), and LTR retrotransposons. All noncoding regions that can be unambiguously associated with a gene (*e.g.*, its UTRs and introns) are merged and treated as one noncoding region to increase the power of our detection of adaptive noncoding regions. Any continuous noncoding region lacking a specific annotation is considered an intergenic noncoding region. We subsequently merged separate, intergenic noncoding regions that are located between two adjacent genes and treated them as one intergenic noncoding region. In our study, we used combined UTR annotations of two previous studies[66,67] and noncoding RNA annotations from a previous study[68]. Other annotations were based on *Saccharomyces* Genome Database[69] (SGD, https://www.yeastgenome.org). The annotations used in this study are provided in Supplementary Data 4.

The relative power in identifying putatively adaptive genes and noncoding regions depends on the quality of annotation of functional units in coding and noncoding regions. While one may think that each gene is naturally a functional unit, this is not necessarily true, because a gene could contain multiple functional units[70] and a functional unit could also be composed of multiple functionally redundant genes[71]. The same can be said of noncoding regions. Hence, we tentatively assume a similar statistical power in identifying putatively adaptive genes and noncoding regions.

## Relative numbers of various types of substitutions under neutrality

Based on the genome annotation from SGD, the number of coding sites relative to the number of noncoding sites in the yeast genome is 2.68, which is taken as the number of coding substitutions relative to that of noncoding substitutions expected under neutrality. To estimate the

relative numbers of synonymous, nonsynonymous, and nonsense substitutions in yeast coding regions expected under neutrality, we used the yeast coding sequence and the mutation spectrum previously estimated from a yeast mutation accumulation study[27]. Specifically, the relative mutation rates for A:T to G:C, A:T to C:G, A:T to T:A, C:G to A:T, C:G to G:C and C:G to T:A were 0.195, 0.110, 0.077, 0.286, 0.067, and 0.265, respectively, when the mutation accumulation data collected from seven different environments were merged. These relative mutation rates became 0.093, 0.221, 0.081, 0.314, 0.093, and 0.198, respectively, when the mutation accumulation data from YPD were used. We also estimated the neutral expectation of the number of frame-shifting indels relative to that of frame-conserving indels using the above mutation data from the seven environments. Because the mutation accumulation data contained fewer than five frame-shifting and frame-conserving indel mutations under YPD, we did not use this dataset to estimate the relative numbers of these indels under neutrality.

## Dice's coefficient of similarity

To assess the similarity in putatively adaptive genes between two evolved lines, we computed Dice's coefficient of similarity[57] by $2|X \cap Y|/(|X|+|Y|)$, where $|X|$ and $|Y|$ represent the number of putatively adaptive genes in the two lines, respectively, and $|X \cap Y|$ represents the number of putatively adaptive genes shared between the two lines. Dice's coefficient ranges from 0 (when the two lines share no putatively adaptive genes) to 1 (when they have the same putatively adaptive genes). For each environment, we calculated the average Dice's coefficient for all pairs of the 12 genomes from the end populations. We also computed the average Dice's coefficient for all pairs of genomes derived from different environments. For comparison, we calculated the expected Dice's coefficient between two genomes if their putatively adaptive genes are independently randomly sampled from all genes in the yeast genome. Note that the gene length variation needed not be considered here because the method used for identifying adaptive genes already ensured that all genes have the same chance probability to be identified as adaptive. We similarly computed Dice's coefficient of similarity in putatively adaptive genes between environments in the same group and those in different groups in Supplementary Fig. 1b, respectively. In Supplementary Fig. 5e, for each of the four environments with 96 replicates, we similarly calculated the average Dice's coefficient of similarity in putatively adaptive genes between samples within the same environment and those in different environments, as well as the expected Dice's coefficient. To examine whether the number of replicates affects Dice's coefficient of similarity in adaptive genes, we randomly downsampled 12 replicates from the 96 replicates in each of the four environments, followed by the inference of putatively adaptive genes based on the 12 replicates. We then computed Dice's coefficient for each pair of samples within and between environments as described earlier. For each environment, we used the mean Dice's coefficient from 100 independent downsamplings as Dice's coefficient from 12 replicates.

## Overall contribution of each adaptive gene

We estimated the overall contribution of each identified adaptive gene to the adaptations in the 252 environments. Let $n$ be the number of identified adaptive genes in an environment. The contribution of a gene to the adaptation in that environment is 0 if the gene is not an adaptive gene in that environment, or $1/n$ if it is an adaptive gene in that environment. The gene's overall contribution is its contribution in each environment averaged across the 252 environments.

## Protein and genetic interactions

The list of protein and genetic interactions in *S. cerevisiae* was obtained from Biological General Repository for Interaction Datasets (BioGRID, https://www.thebiogrid.org), version 4.4.198, which contained 176,516 protein-protein and 584,211 genetic interactions, respectively[72].

## Gene Ontology (GO) analysis

We used the program GO Term Finder[73], available from the SGD[69], to identify biological process, molecular function, and cellular component terms enriched in the 149 identified adaptive genes. *P*-values were adjusted using the Bonferroni correction for multiple testing.

## Other data sources

The fitness effects of yeast gene deletions under SC (Fig. 4a) were from Qian et al. [29]. whereas the mRNA expression levels of yeast genes in rich medium (Fig. 4b) were from Chou et al. [74].

## Estimation of ω

We used the CODEML program from the PAML package[75] to estimate the $\omega$ values of *S. cerevisiae* and *S. paradoxus* orthologous genes. Each gene analyzed must fulfill three criteria: (1) it has one-to-one orthologs in the two species, (2) the *S. cerevisiae* ortholog must have a verified open reading frame (ORF) instead of a dubious ORF, uncharacterized ORF, or pseudogene, (3) the *S. paradoxus* ortholog must not contain premature stop codons or frame-shifting indels when compared with the *S. cerevisiae* sequence. In total, 4716 pairs of orthologous sequences were included.

## Introducing putatively adaptive substitutions into the progenitor

We chose one observed substitution from each of eight commonly used adaptive genes and introduced it into the progenitor strain using CRISPR/Cas9 genome editing. All primers used are provided in Supplementary Data 6. The donor DNA for CRISPR/Cas9 editing was generated by annealing and extending two primers containing the targeted mutation. The guide RNA (gRNA) was produced by annealing two primers and ligating the product to a backbone plasmid (pML104-KanMX4) with a Cas9 expression module, following a previous protocol[76]. The resulting plasmid was confirmed by Sanger sequencing. The donor DNA and the gRNA plasmid were introduced into cells of the progenitor strain by high-efficiency transformation[77]. The KanMX4 gene on the plasmid provided yeast with resistance to geneticin, allowing selection for correct genotypes after yeast transformations. Colonies were picked from selection plates and the intended genetic changes were subsequently confirmed by Sanger sequencing. The Cas9 plasmids for all mutants were eliminated through serial transfers in the SC medium every 12 h for a total of 36 h. Colonies lacking Cas9 plasmids were confirmed by plating cells on plates with geneticin. We then measured the maximum growth rate ($m_i$) of each mutant in ten home environments and ten foreign environments, performing three replicate measurements for each. In each relevant environment, we also measured the mean maximum growth rate of the progenitor from three replicates ($\overline{R_i}$). The fitness of a mutant relative to the progenitor in environment $i$ was estimated by $2^{m_i/\overline{R_i}-1}$. The reported relative fitness represents the mean ± standard error from the three replicates.

## Fraction of the fitness gain that occurred in the first 800 generations of adaptation

Johnson et al. used three yeast strains to perform laboratory adaptation for 10,000 generations in each of three environments[48]. Their fitness data collected at the 70th, 550th, and 10,000th generations showed that the median fraction of the total fitness gain in the experiment attributable to the time between the 70th and 550th generations is 23.5%. If we assume that the rate of the fitness gain per generation in the first 800 generations is the same as that from the 70th to 550th generations, the fraction of the total fitness gain that occurred in the first 800 generations is 39.2%.

## Reporting summary

Further information on research design is available in the Nature Portfolio Reporting Summary linked to this article.

## Data availability

The Illumina sequencing data have been deposited to NCBI SRA under the accession number PRJNA1019277. The list of protein and genetic interactions in *S. cerevisiae* was obtained from BioGRID (https://www.thebiogrid.org), version 4.4.198. The fitness effects of yeast gene deletions under SC were from ref. 29. whereas the mRNA expression levels of yeast genes in rich medium were from ref. 74. Source data are provided as a Source Data file. Source data are provided with this paper.

## Code availability

Custom code can be downloaded from Github (https://github.com/PiaopiaoChen/Environmental_adaptations.git) or Zenodo (https://doi.org/10.5281/zenodo.11420291).

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

## Acknowledgements

We thank Vivian Yan for technical assistance and H. Liu, A. Mahilkar, S. Song, and H. Xu for valuable comments. This work was supported by the U.S. National Institutes of Health grant R35GM139484 to J.Z.

## Author contributions

J.Z. conceived of the project and secured funding; P.C. and J.Z. designed the study; P.C. performed the experiments and analyzed the data; P.C. and J.Z. wrote the paper.

## Competing interests

The authors declare no competing interests.
