## [Peer Review File · Nature Communications]

The loci of environmental adaptation in a model eukaryoteREVIEWER COMMENTS

Reviewer #1 (Remarks to the Author):

The authors explore the adaptation capabilities of yeast in various laboratory conditions, beginning with a single diploid yeast strain and propagated it across 3,024 populations for 800 generations. The authors pre-adapted the progenitor yeast strain to synthetic complete (SC) medium for 600 generations before initiating the experiment. This was done in 252 distinct laboratory environments, each environment being represented by 12 replicated cultures.

This study represents a clearly written, comprehensive and systematic approach to understanding yeast evolution in controlled conditions, with a significant scale that could provide insights into the genetic basis of adaptation and the potential for yeast to cope with diverse and changing environments.

This study is already of a high quality, and will be a valuable contribution to the literature, I just have a few comments.

1. The initial results section that summarises the changes of fitness is beautifully written and the corresponding figure one is a marvel to look at. If I have one note it could be to extend your explanation of what is going on in figure 1C just a little – it is such a cool result and it would be good to draw the attention of the nonspecialist reader to this.

2. Foreign vs home environments. The main methods needs a clear explanation for how these are defined, I could not figure it out.

3. Working out which genes are adaptive and Dices coefficient – how do you generate your null-expectation of the similarity given that mutational processes are not completely random - or example larger genes are more likely to have mutations in small genes. Also, mutation is often modelled as a poisson process, how did you “randomly sample from all genes”. Methods need some more detail and explanation.

4. I really like that this paper is written from the perspective of someone with specialist knowledge of *Drosophila* comparative genomics, but there are a couple of experimental studies of yeast populations (usually in a single glucose rich media) that should be cited. Moreover, I think it would enrich the discussion if the authors incorporated some of this context when explaining their results.

First Lang et al 2013 (DOI: 10.1038/nature12344) and Sherlock et al 2016 (doi: 10.1016/j.cell.2016.08.002) identified IRA2 and PDR1 and other parts of the RAS pathway as adaptive targets in experimental evolution, with Sherlock having an extensive discussion on the RAS pathway.

5. The authors write that the fitness of key mutations is larger in growth conditions with higher stress levels. However, they do not measure stress levels, rather they assume levels of stress based on how well the strain performs in a different environment/growth conditions. This means that the difference in fitness that they observe is actually corresponding to a change in fitness, or environment, rather than strictly “stress level”. I would like the authors to justify the use of the term stress level and explain why it is more appropriate than writing that the fitness effects of genetic variants is greater in environments where the background genotype has a low fitness.

6. There have been a few papers that have identified a very similar trend when measuring the fitness effects of a genetic variant engineered into different genetic backgrounds- referred to as “diminishing returns epistasis” discussed extensively in these articles DOI: 10.1126/science.1203799 and DOI: 10.1126/science.1203801. In particular Kryazhimskiy et al 2015 did a reconstruction experiment with Whi2 and other genes (doi: 10.1126/science.1250939). Clearly these phenomena have a shared principle of relative benefit: that the relative benefit of a mutation is context-dependent. I think the authors need to point out this other work, and then, if they like, to discuss any potential generalities that arise from a synthesis of their results and

others.

Notes on figures.

Data availability – I could not access the data using the PRJNA number (not just checking the link, also following up in bioprojects) and the github link did not work. The text says that the 252 environments are available in Data S1 – while there are figures that show the classes of environments, I could not find a list of the environments. That's three out of three not working! Given that a major strength of this study is the data set, I will need to confirm that data are available before I could support publication of this paper.

Reviewer #1 (Remarks on code availability):

github link did not work 404 error

Reviewer #2 (Remarks to the Author):

Chen et al,

Chen and Zhang artificially adapted a diploid yeast lab strain to many simple environments and short-read sequenced the start and many endpoint populations. This allows them to address general questions on the genetics of adaptation, such what type of genes putatively adaptive mutations tend to occur in and whether they affect coding or non-coding sites. One can discuss how broadly these findings can be extrapolated from simple to more complex natural environments and outside of this particular genetic context. To some extent the authors also do this in a supplementary note. And the authors do not consider the dynamics of adaptation or evolution. But, the view provided here is remarkably broad, and can be expected to serve as a reference point for future papers in this area. Overall, I am therefore favourably disposed towards this paper and find it suitable for Nature Communications.

1. My main beef with this submission is the low resolution of the environmental dimension. After having introduced and categorized environments in Fig S1, the authors mostly fail to use this information further. Instead, they treat the environmental dimension as a black box for most of the paper, with the environmental axes mostly just indicating "252 environments". Given that the environments seem to be intentionally chosen to fall into meaningful categories, and many having quite obvious links to the putatively adaptive genes emerging in the screen, this is a pity. For example, beneficial mutations in the RAS/PKA pathway often emerge when growth is limited by carbon/energy access, mutations in PDR genes are known to drive adaptation to many toxic compounds, mutations in nitrogen transport and metabolism could be suspected of driving adaptation to nitrogen restricted environments, mutations in metal metabolism could be expected to drive adaptation to alterations in metal concentrations, mutagenic agents (not only 4NQO) could be expected to increase the substitution rate. And yet, despite environments falling into obvious such categories, the authors make few attempts to address links of this type – although they should have a quite unique power to do so. Moreover, they do little effort to connect the results to previous experimental evolution studies using the same, or very similar, selective pressures. The absence of a deeper analysis of gene-environment and mutation-environment relations unnecessarily limits our understanding of the genetic patterns reported in the paper. Rather than being general phenomena, these may very well be connected to subsets of the selection pressures used.

2. From a methodological perspective, I wasn't entirely convinced by how the authors express in text what they measure - and by how they arrive at these estimates. The authors make a point out of stating that they measure fitness, rather than a phenotype associated with fitness. But what they do measure are growth rates - which is a phenotype associated with fitness. Fitness is a complex property that involves effects on other growth properties (e.g. lag, yield) – and in this experimental context probably also properties such as buoyancy, sedimentation, aggregation, adherence, etc. Using "fitness", "adaptation" or "growth" without the growth rate qualifier is thus misleading. Second, the growth rates were measured using a 96 well shaker/optical reader, and while I am not familiar with this brand, it would be important to correct for the non-linearity

between optical density and population size to get biologically meaningful growth rates. It wasn't clear to me if or how this normalization was done. Third, the authors are doing experimental evolution with serial batch transfers of a fixed volume of medium after 24h, at which point populations are said to be in stationary phase. But, the many environments affect growth lag, rate and/or yield differently. Moreover, these are probably all evolving phenotypes that are under positive selection. This means that not all populations, at all time points, will have reached stationary phase after 24h. And even if they did, not all stationary phase populations would have the same cell concentrations because the stationary phase yield would differ. This variation will affect the estimates of number of cell generations very substantially. The rather exact number of 800 generations given will therefore, for many environments, be quite incorrect. The authors are not considering the dynamics of adaptation in this paper, so the error is not grave - but they should include a more honest estimate of number of generations of adaptation.

3. At various points, the authors consider the negative correlation between selection strength and adaptive gain. This slower adaptation for less stressed populations is very expected. In fact, many papers have reported just that for genotypes that experience different degrees of intracellular stress due to sub-optimal genome composition. In fact, the predictive power of stress level on adaptation appear to be quite outstanding (with PMID: 36083011 being a recent paper). I think the obvious parallels here deserve a more extensive discussion. From the perspective of the cell, the distinction of whether intracellular stress is caused by genotype, genotype-by-environment interactions, or environment only - may be entirely trivial. With this in mind, I am more curious about the outliers in Fig 1C that deviate from this expected negative correlation - i.e. the environments where selection is strong but next to no adaptation is achieved. Is this biology - or just a lack of measurement precision? In absence of error bars, this is not easy to deduce. If biology, how do we explain this absence of adaptation? For unfit genotypes, there are virtually no such strong outliers - they all manage to adapt fast (see e.g. the ref above). From a more general perspective, the information content of the scatter plots in Fig 1 and S2 is unnecessarily low - it would help e.g. if the dots were coloured according to environment type and outliers were indicated with names.

4. I appreciate the honest discussion in the supplementary note about the limitations of the study and the degree to which caution should be employed when extrapolating the reported findings. I think it would be prudent to modify the statement that the external environment. As these are batch growth, rather than chemostat, experiments, the expanding cell populations are continuously modifying the external environment by uptake and secretion of compounds, in particular changing the nutrient content of the environment. It would probably also be correct to recognize that natural environments are much more complex in terms of the number and type of selection pressures that manifest simultaneously - and in terms of yeast these selection pressures would act also on aspects of the life cycle other than mitotic growth. The lab strain used by the authors are also a remarkably poor representative of yeast as a species, and in particular of non-domesticated yeast, with well-known defects in e.g. mitochondrial biology. To me, it is far from clear that the findings reported here would hold true from natural yeast strains, and the authors arguments that they should to be would need further clarification. Finally, the types of mutations and genes that drive adaptation can be expected to depend on the interplay between mutation rates, population size and selection strength (e.g. PMID: 20308101). The population size in this experimental set-up is remarkably small relative natural yeast populations. This could e.g. make it hard for populations to encounter, within the time frame here considered, beneficial mutations in short genes (for which the mutation target is smaller), beneficial gain-of-function mutations (which should be rarer than corresponding loss-of-function mutations) or other rare beneficial mutation types (most structural variations) - and thus influence the generality of the results.

Minor comments:

5. Row 183: "Frame-shifting insertion/deletion (indel) mutations in coding regions are expected to be under much stronger negative selection than are frame-conserving indels." This probably holds true in natural environments for virtually all genes, because purifying selection acting to retain protein function unchanged probably dominates completely. But in any of the simple environments explored, loss-of-function may be beneficial for a substantial number of genes and thus loss-of-function mutations, e.g. frame-shifting indels, may in many genes not be under stronger negative

selection.

6. Row 205. "Notably, six of the seven adaptive ARSs were identified from the 4-NQO environment, suggesting the possibility that ARS mutations mitigate the inhibition of replication by 4-NQO". I found this a very interesting finding. Do these six ARS correspond to different plasmids, or different segments of the same plasmid? How do the authors envision that the beneficial effect arises – through mutations affecting the origin of replication which increases the plasmid copy number of a resistance elements carrying plasmids?

Reviewer #2 (Remarks on code availability):

Im not qualified to evaluate the code

Reviewer #3 (Remarks to the Author):

This is a well-written manuscript detailing a nice study which documents 800 generations of laboratory evolution in yeast across 252 unique environments. The diversity of environments explored here make this study stand out as novel and impactful. This study attempts to synthesize common threads emerging from these different evolutionary treatments. The authors report that most putatively adaptive alleles are large-effect mutations in a set of "adaptation genes" that are implicated repeatedly. I think this paper has the potential to be highly cited in the field of experimental evolutionary genomics, and I enjoyed reading it. I do have some constructive feedback that the authors may want to consider during the revision process.

Major comments

1. The yeast genome mostly consists of protein-coding genes. The authors allude to this in the text, citing the Supplementary Note 1, though that note is a general list of the study's limitations and doesn't expand on this specific point much. While I am convinced that large-effect mutations in coding regions are common during adaptation in yeast, this does call into question whether this might hold true for higher eukaryotes. Some more recognition/speculation around this point would provide some balance.
2. Section beginning 211 – the adaptive genes section. I would love to see more synthesis/meta-analysis of the degree to which these genes are implicated in other experimental evolution studies. This point also applies to the section beginning line 641 (which does not include comparison of present results to Johnson et al. 2021)
3. The main phenotype measured in this study is growth rate relative to the ancestor. While the inclusion of functional validation of putatively adaptive alleles via CRISPR is very impressive, I think it also fair to point out that growth rate is not the only predictor of fitness in yeast (though it is certainly the easiest to measure in high-throughput). At the very least, I'd expect to see some acknowledgement of this point somewhere in the manuscript, and perhaps this could be included in the Supplementary Note that enumerates some of the study's limitations.
4. I would have valued more discussion of within-treatment variation. Clearly, the point here is to identify adaptive alleles/genes that are shared across treatments, but your data may shed light on the question of whether parallel evolution across replicates might be environment-specific. Related to this point – you define 2 general treatment classes in this study (chemicals vs. more "natural" stressors). It wasn't obvious why these two classes were designated, and the degree to which they may have been analyzed separately. For example, the PDR genes; these are ABC transporters that are expected to evolve in response to chemical/drug stress. PDR3, PDR1, and PDR5 are all included in the list of "general adaptation genes", but is this due to repeatedly showing up in the chemical stress treatments, or did they indeed show up across all treatments? Regarding the fitness data, it would also be interesting to see if the fitness changes were greater overall in one treatment class versus another. Perhaps this could be incorporated into Figure 1a, with some type of color coding to show those environments belonging to each class.

Minor comments

Line 83: "this enrichment may be attributable to a higher power of the approach in detecting diversifying selection than directional selection instead of an oversized contribution of these genes to adaptation". This sentiment is not clear to me as written, can you rephrase? I don't think of directional selection as difficult to detect, just rarely detected.

Throughout the manuscript "adaptations" is often referred to as plural when singular works better (for example, in the title).

First sentence in Discussion is too speculative. According to my understanding, this study does not show that there is an increasing beneficial mutation rate with stress level, there is just a literature precedent to support this point. The study does test the idea that the effect of a mutation increases with stress level so that part is OK.

Lines 160-161: I understand what is being stated here, but the language could be clearer. I think this statement could be clarified by simply stating that the higher substitution rate suggests stronger positive selection on coding regions in the observed data.

Line 343: "agnostic" works better than "agonistic", typo?

Lines 353: appear to*

Line 425: the "Experimental evolution" section of the methods seems a bit cursory. For example, was there some liquid handling automation (I would think this is likely given the high throughput of the experiment)? If so this is not mentioned.

Line 435: "we performed experimental evolution of 84 additional replicates in each of four environments chosen from the 252 environments"....to assess the impact of # reps on Dice's coefficient of similarity. I am not finding much discussion of this part of the experiment in the main text. This is related to major concern #4 – you seem to have measured the degree of evolutionary parallelism across replicates at high depth (at least for 4 treatments), but do not focus on that outcome.

RESPONSE TO REVIEWERS' COMMENTS

We thank the three reviewers for their valuable comments, which have helped improve our manuscript. Below please find our point-to-point response in blue.

Reviewer #1:

The authors explore the adaptation capabilities of yeast in various laboratory conditions, beginning with a single diploid yeast strain and propagated it across 3,024 populations for 800 generations. the authors pre-adapted the progenitor yeast strain to synthetic complete (SC) medium for 600 generations before initiating the experiment. This was done in 252 distinct laboratory environments, each environment being represented by 12 replicated cultures.

This study represents a clearly written, comprehensive and systematic approach to understanding yeast evolution in controlled conditions, with a significant scale that could provide insights into the genetic basis of adaptation and the potential for yeast to cope with diverse and changing environments.

This study is already of a high quality, and will be a valuable contribution to the literature, I just have a few comments.

We thank the reviewer for the very positive evaluation. We address the specific comments below.

1. The initial results section that summarises the changes of fitness is beautifully written and the corresponding figure one is a marvel to look at. If I have one note it could be to extend your explanation of what is going on in figure 1C just a little – it is such a cool result and it would be good to draw the attention of the nonspecialist reader to this.

We appreciate the kind words on this section. Following the suggestion, we have added a mechanistic explanation of the observation made in Fig. 1c as follows (lines 132-133).

“In theory, this trend could be caused by an increase in the beneficial mutation rate and/or size with the level of environmental stress (see below).”

2. Foreign vs home environments. The main methods need a clear explanation for how these are defined, I could not figure it out.

In our study, the term “home environments” refers to the environments where a focal gene was identified as adaptive. Conversely, “foreign environments” are those where no substitution in the gene was observed. It is important to note that the home/foreign classification is specific to each gene; different genes have different home and foreign environments. We randomly selected ten

home and ten foreign environments for each gene to evaluate the fitness effects of certain substitutions in these genes. We have now added the following sentences to clarify these points (lines 290-293).

“For each gene of interest, we randomly chose ten “home” environments, where the gene was identified as adaptive, and ten “foreign” environments, where no substitution was observed in the gene. Note that different genes typically have different home/foreign environments.”

3. Working out which genes are adaptive and Dices coefficient – how do you generate your null-expectation of the similarity given that mutational processes are not completely random - or example larger genes are more likely to have mutations in small genes. Also, mutation is often modelled as a Poisson process, how did you “randomly sample from all genes”. Methods need some more detail and explanation.

We considered gene length in the identification of adaptive genes. As described in lines 533-543, in each sample, we randomly generated the same number of substitutions in coding regions as observed in the sample. This means that a longer gene had a proportionately higher mean number of random substitutions. Consequently, all genes have an equal probability to be identified as adaptive by chance. To calculate the null expectation for Dice's coefficient, which compares the similarity of adaptive genes across samples, in each sample we randomly picked from all genes in the yeast genome the same number of genes as the number of observed adaptive genes in the sample. We have provided additional explanations on these points in Methods (line 538 and 600-603).

“Thus, longer genes had proportionately more random substitutions.”

“Note that the gene length variation needed not be considered here because the method used for identifying adaptive genes already ensured that all genes have the same chance probability to be identified as adaptive.”

4. I really like that this paper is written from the perspective of someone with specialist knowledge of Drosophila comparative genomic's, but there are a couple of experimental studies of yeast populations (usually in a single glucose rich media) that should be cited. Moreover, I think it would enrich the discussion if the authors incorporated some of this context when explaining their results.

First Lang et al 2013 (DOI: 10.1038/nature12344) and Sherlock et al 2016 (doi-10.1016/j.cell.2016.08.002) identified IRA2 and PDR1 and other parts of the RAS pathway as adaptive targets in experimental evolution, with Sherlock having an expensive discussion on the RAS pathway.

We thank the reviewer for pointing us to these references, which have now been cited (lines 386-395).

5. The authors write that the fitness of key mutations is larger in growth conditions with higher stress levels. However, they do not measure stress levels, rather they assume levels of stress based on how well the strain performs in a different environment/growth conditions. This means that the difference in fitness that they observe is actually corresponding to a change in fitness, or environment, rather than strictly “stress level”. I would like the authors to justify the use of the term stress level and explain why it is more appropriate than writing that the fitness effects of genetic variants is greater in environments where the background genotype has a low fitness.

We thank the reviewer for this suggestion and have added the following justification to the main text (lines 116-119).

“Given that we evolved the same progenitor across all environments, the fitness of the progenitor under an environment reflects the degree of environmental challenge to the strain so can be used to measure the environmental stress level, with a lower fitness indicating a higher level of environmental stress.”

6. There have been a few papers that have identified a very similar trend when measuring the fitness effects of a genetic variant engineered into different genetic backgrounds- referred to as “diminishing returns epistasis” discussed extensively in these articles DOI: 10.1126/science.1203799 and DOI: 10.1126/science.1203801. In particular Kryazhimskiy et al 2015 did a reconstruction experiment with Whi2 and other genes (doi: 10.1126/science.1250939). Clearly these phenomena have a shared principle of relative benefit: that the relative benefit of a mutation is context-dependent. I think the authors need to point out this other work, and then, if they like, to discuss any potential generalities that arise from a synthesis of their results and others.

The studies mentioned by the reviewer focused on the effect of the same mutation across different genetic backgrounds under the same environment, whereas our work investigates the impact of the same mutation in the same genetic background across different environments. We discussed the similarity and differences between the two as follows (lines 346-351).

“The latter trend is analogous to the widely reported trend that the rate of fitness increase of an adapting population declines as the population adapts to its environment, owing to diminishing returns epistasis^{13,14,16,31-35}. These previous studies, along with the present research, demonstrate the context-dependency of the fitness effects of mutations, where the previous studies focused on the genetic context while our study focused on the environmental context.”

Data availability – I could not access the data using the PRJNA number (not just checking the

link, also following up in bioprojects) and the github link did not work. The text says that the 252 environments are available in Data S1 – while there are figures that show the classes of environments, I could not find a list of the environments. That's three out of three not working! Given that a major strength of this study is the data set, I will need to confirm that data are available before I could support publication of this paper.

Reviewer #1 (Remarks on code availability):

github link did not work 404 error

The link to the BioProject (and associated SRA metadata) is as follows:

<https://dataview.ncbi.nlm.nih.gov/object/PRJNA1019277?reviewer=d114hspfoco01dnd55n3r46i0p>. We will make all data publicly available immediately upon the acceptance of the manuscript. Additionally, we have made the GitHub link public (https://github.com/PiaopiaoChen/Environmental_adaptations.git). In Data S1, the third column presents the name of each environment, detailing the specific ingredients added or removed. Information regarding the base media is provided in the last column.

Reviewer #2:

Chen and Zhang artificially adapted a diploid yeast lab strain to many simple environments and short-read sequenced the start and many endpoint populations. This allows them to address general questions on the genetics of adaptation, such what type of genes putatively adaptive mutations tend to occur in and whether they affect coding or non-coding sites. One can discuss how broadly these findings can be extrapolated from simple to more complex natural environments and outside of this particular genetic context. To some extent the authors also do this in a supplementary note. And the authors do not consider the dynamics of adaptation or evolution. But, the view provided here is remarkably broad, and can be expected to serve as a reference point for future papers in this area. Overall, I am therefore favourably disposed towards this paper and find it suitable for Nature Communications.

We thank the reviewer for the very positive evaluation. We address the specific comments below.

1. My main beef with this submission is the low resolution of the environmental dimension. After having introduced and categorized environments in Fig S1, the authors mostly fail to use this information further. Instead, they treat the environmental dimension as a black box for most of the paper, with the environmental axes mostly just indicating “252 environments”. Given that the environments seem to be intentionally chosen to fall into meaningful categories, and many having quite obvious links to the putatively adaptive genes emerging in the screen, this is a pity. For example, beneficial mutations in the RAS/PKA pathway often emerge when growth is limited by carbon/energy access, mutations in PDR genes are known to drive adaptation to many toxic compounds, mutations in nitrogen transport and metabolism could be suspected of driving adaptation to nitrogen restricted environments, mutations in metal metabolism could be expected

to drive adaptation to alterations in metal concentrations, mutagenic agents (not only 4NQO) could be expected to increase the substitution rate. And yet, despite environments falling into obvious such categories, the authors make few attempts to address links of this type – although they should have a quite unique power to do so. Moreover, they do little effort to connect the results to previous experimental evolution studies using the same, or very similar, selective pressures. The absence of a deeper analysis of gene-environment and mutation-environment relations unnecessarily limits our understanding of the genetic patterns reported in the paper. Rather than being general phenomena, these may very well be connected to subsets of the selection pressures used.

We appreciate this constructive comment. For three main reasons, we did not extensively compare the adaptive genes across classes/groups of environments. First, the classification of environments into two broad classes was somewhat arbitrary. This is especially true to Class II environments, because they are a heterogenous set of chemicals that were further divided into 10 groups based on the fitness profiling of approximately 5,900 yeast gene deletion strains. Even within the same group, the different environments could be quite different. Second, we found no significant difference in the mean Dice's coefficient of similarity in putatively adaptive genes between different environments within a group and those between groups (Fig. S5c), suggesting that the adaptive genetic responses observed are not substantially different among groups of environments. Finally, the primary focus of the study is to elucidate general patterns in the genomic basis of adaptation rather than delving into the specifics of which genes are utilized in response to which environment. Notwithstanding, we followed the reviewer's suggestion to analyze whether there are differences in adaptive genes across classes/groups of environments. The results have been added to the main text (lines 275-285), as shown below.

“Finally, we compared the adaptive genes identified in the two large classes of environments used in our experimental evolution (Fig. S1). We observed a higher prevalence of PDR genes (PDR1, PDR3, and PDR5) in Class II environments, whereas signal transduction genes such as IRA2, CYR1, and RAS2 are enriched in Class I environments. Specifically, 34% (52) of Class II environments but only 3% (3) of Class I environments used PDR genes ($P = 1.1 \times 10^{-7}$, chi-squared test). Conversely, 66% of Class I environments but only 52% of Class II environments used signal transduction genes ($P = 0.065$). Further subdivision of Class I environments into five groups (Fig. S1b) revealed a propensity for transmembrane genes in environments with different nitrogen resources, notably GNP1. Specifically, 41% (9) of environments with different nitrogen resources utilized GNP1, compared to only 15% in all other environments studied ($P = 0.0052$).”

2. From a methodological perspective, I wasn't entirely convinced by how the authors express in text what they measure - and by how they arrive at these estimates. The authors make a point out of stating that they measure fitness, rather than a phenotype associated with fitness. But what they do measure are growth rates - which is a phenotype associated with fitness. Fitness is a complex property that involves effects on other growth properties (e.g. lag, yield) – and in this

experimental context probably also properties such as buoyancy, sedimentation, aggregation, adherence, etc. Using “fitness”, “adaptation” or “growth” without the growth rate qualifier is thus misleading. Second, the growth rates were measured using a 96 well shaker/optical reader, and while I am not familiar with this brand, it would be important to correct for the non-linearity between optical density and population size to get biologically meaningful growth rates. It wasn't clear to me if or how this normalization was done. Third, the authors are doing experimental evolution with serial batch transfers of a fixed volume of medium after 24h, at which point populations are said to be in stationary phase. But, the many environments affect growth lag, rate and/or yield differently. Moreover, these are probably all evolving phenotypes that are under positive selection. This means that not all populations, at all time points, will have reached stationary phase after 24h. And even if they did, not all stationary phase populations would have the same cell concentrations because the stationary phase yield would differ. This variation will affect the estimates of number of cell generations very substantially. The rather exact number of 800 generations given will therefore, for many environments, be quite incorrect. The authors are not considering the dynamics of adaptation in this paper, so the error is not grave - but they should include a more honest estimate of number of generations of adaptation.

The reviewer asked three methodological questions. First, we agree with the reviewer that lag time, maximum growth rate in the log phase, and yield are all relevant components of fitness. In this work, we focused on maximum growth rate and used it as a proxy for fitness. This is now clarified (lines 114-116).

Second, we converted OD to cell density using a pre-determined OD-cell density curve. We are sorry that this step was not described in our original manuscript. It has now been added in Methods (lines 470-473).

Third, we adjusted chemical concentrations in the media to ensure that the doubling time for the progenitor in each environment is shorter than three hours and that population growth reached the stationary phase in 24 hours. By transferring 2 μ l of culture into 500 μ l of fresh culture medium every 24 hours, we ensured that the culture underwent $\log_2 250 \approx 8$ generations. Hence, the number of generations was controlled. We have clarified this point in Methods (lines 437-439 and 451-454).

3. At various points, the authors consider the negative correlation between selection strength and adaptive gain. This slower adaptation for less stressed populations is very expected. In fact, many papers have reported just that for genotypes that experience different degrees of intracellular stress due to sub-optimal genome composition. In fact, the predictive power of stress level on adaptation appear to be quite outstanding (with PMID: 36083011 being a recent paper). I think the obvious parallels here deserve a more extensive discussion. From the perspective of the cell, the distinction of whether intracellular stress is caused by genotype, genotype-by-environment interactions, or environment only - may be entirely trivial. With this in

mind, I am more curious about the outliers in Fig 1C that deviate from this expected negative correlation – i.e. the environments where selection is strong but next to no adaptation is achieved. Is this biology - or just a lack of measurement precision? In absence of error bars, this is not easy to deduce. If biology, how do we explain this absence of adaptation? For unfit genotypes, there are virtually no such strong outliers – they all manage to adapt fast (see e.g. the ref above). From a more general perspective, the information content of the scatter plots in Fig 1 and S2 is unnecessarily low – it would help e.g. if the dots were coloured according to environment type and outliers were indicated with names.

We have now discussed the similarities and differences between our study and the one mentioned by the reviewer (and several other papers) as follows (lines 346-351):

“The latter trend is analogous to the widely reported trend that the rate of fitness increase of an adapting population declines as the population adapts to its environment, owing to diminishing returns epistasis^{13,14,16,31-35}. These previous studies, along with the present research, demonstrate the context-dependency of the fitness effects of mutations, where the previous studies focused on the genetic context while our study focused on the environmental context.”

In our study, the distinction in intracellular stress is exclusively due to environment factors because we used the same ancestral strain in all experimental evolution. In fact, we found that the extent of adaptation (f_i/F_i) exceeded 1 across all environments (the y-axis values in Fig. 1c are all above 1). Notably, 95.7% of cases showed significant adaptation, whereas 4.3% did not exhibit significant adaptation. The non-significant adaptation could be due to growth rate measurement precision issues or influences on other fitness components such as lag time or yield, which were not measured. Given that Fig. 1c underscores the negative correlation between the relative progenitor fitness in an environment and the extent of adaptation in the environment, adding colors or names to mark the environments could potentially cause distraction from the primary observation. Nevertheless, we have taken the suggestion to produce such a figure below.

Additionally, we added colors to Fig. 1ab and included them as Fig. S2 (see also below). However, we did not find any apparent patterns. For example, the extent of adaptation (f_i/F_i) is not significantly different between the two classes of environments ($P = 0.47$, two-tailed Wilcoxon rank-sum test).

4. I appreciate the honest discussion in the supplementary note about the limitations of the study and the degree to which caution should be employed when extrapolating the reported findings. I think it would be prudent to modify the statement that the external environment. As these are batch growth, rather than chemostat, experiments, the expanding cell populations are continuously modifying the external environment by uptake and secretion of compounds, in particular changing the nutrient content of the environment. It would probably also be correct to recognize that natural environments are much more complex in terms of the number and type of selection pressures that manifest simultaneously – and in terms of yeast these selection pressures would act also on aspects of the life cycle other than mitotic growth. The lab strain used by the authors are also a remarkably poor representative of yeast as a species, and in particular of non-domesticated yeast, with well-known defects in e.g. mitochondrial biology. To me, it is far from clear that the findings reported here would hold true from natural yeast strains, and the authors

arguments that they should to be would need further clarification. Finally, the types of mutations and genes that drive adaptation can be expected to depend on the interplay between mutation rates, population size and selection strength (e.g. PMID: 20308101). The population size in this experimental set-up is remarkably small relative natural yeast populations. This could e.g. make it hard for populations to encounter, within the time frame here considered, beneficial mutations in short genes (for which the mutation target is smaller), beneficial gain-of-function mutations (which should be rarer than corresponding loss-of-function mutations) or other rare beneficial mutation types (most structural variations) – and thus influence the generality of the results.

In Supplementary Note 1, we have now expanded the discussion of the limitations of the present study including, for example, sexual/asexual reproduction, changing environments, and population size that could potentially affect the generalizability of our findings. However, we chose not to discuss chemostats vs. batch cultures because it is unclear which better mimics the complex and fluctuating natural environments of yeast. While we agree that the selection pressures in batch cultures varied over the growth cycle, which could lead to different adaptive outcomes when compared to evolution in the constant environment of a chemostat, this may not be a limitation of our study because the natural environment of yeast could be varying and affected by yeast growth as well.

Minor comments:

5. Row 183: “Frame-shifting insertion/deletion (indel) mutations in coding regions are expected to be under much stronger negative selection than are frame-conserving indels.” This probably holds true in natural environments for virtually all genes, because purifying selection acting to retain protein function unchanged probably dominates completely. But in any of the simple environments explored, loss-of-function may be beneficial for a substantial number of genes and thus loss-of-function mutations, e.g. frame-shifting indels, may in many genes not be under stronger negative selection.

One can consider three situations. First, when the functional of a gene is useful to cell growth, frame-shifting indels should be under stronger purifying selection than frame-conserving indels. Second, when the gene function is useless, the two types of indels would both be neutral. Third, when the gene function is harmful to cell growth, frame-shifting indels might be under stronger positive selection than frame-conserving indels. Even under the experimental evolution setting, most genes should fall into the first situation while a minority of genes may belong to the second or third situation. Hence, frame-shifting indels are overall under stronger purifying selection than frame-conserving indels in natural as well as experimental evolution.

6. Row 205. “Notably, six of the seven adaptive ARSs were identified from the 4-NQO environment, suggesting the possibility that ARS mutations mitigate the inhibition of replication by 4-NQO”. I found this a very interesting finding. Do these six ARS correspond to

different plasmids, or different segments of the same plasmid? How do the authors envision that the beneficial effect arises – through mutations affecting the origin of replication which increases the plasmid copy number of a resistance elements carrying plasmids?

To clarify, ARS refers to autonomously replicating sequences within the yeast nuclear genome, not on plasmids (now clarified in line 205). The adaptive ARSs we identified are distinct elements within the nuclear DNA that contribute to replication initiation.

Reviewer #2 (Remarks on code availability): I'm not qualified to evaluate the code

We have made the GitHub link (https://github.com/PiaopiaoChen/Environmental_adaptations.git) public.

Reviewer #3:

This is a well-written manuscript detailing a nice study which documents 800 generations of laboratory evolution in yeast across 252 unique environments. The diversity of environments explored here make this study stand out as novel and impactful. This study attempts to synthesize common threads emerging from these different evolutionary treatments. The authors report that most putatively adaptive alleles are large-effect mutations in a set of “adaptation genes” that are implicated repeatedly. I think this paper has the potential to be highly cited in the field of experimental evolutionary genomics, and I enjoyed reading it. I do have some constructive feedback that the authors may want to consider during the revision process.

We thank the reviewer for the very positive evaluation. We address the specific comments below.

Major comments

1. The yeast genome mostly consists of protein-coding genes. The authors allude to this in the text, citing the Supplementary Note 1, though that note is a general list of the study's limitations and doesn't expand on this specific point much. While I am convinced that large-effect mutations in coding regions are common during adaptation in yeast, this does call into question whether this might hold true for higher eukaryotes. Some more recognition/speculation around this point would provide some balance.

Given that the genomes of many multicellular eukaryotes typically feature a greater proportion of noncoding regions, the role of noncoding mutations in adaptation may differ from that observed in yeast. We have discussed this point in the Supplementary Note 1 as follows.

“Most importantly, it is unclear whether our conclusions from a unicellular eukaryote are applicable to multicellular eukaryotes, which often possess a much larger fraction of the

noncoding genome.”

2. Section beginning 211 – the adaptive genes section. I would love to see more synthesis/meta-analysis of the degree to which these genes are implicated in other experimental evolution studies. This point also applies to the section beginning line 641 (which does not include comparison of present results to Johnson et al. 2021)

Following the suggestion, we have now added this point and some references in Discussion.

“Some of these adaptive genes such as IRA2, PDR5, and WHI2 were also discovered in previous yeast evolution experiments^{46-48.}”

3. The main phenotype measured in this study is growth rate relative to the ancestor. While the inclusion of functional validation of putatively adaptive alleles via CRISPR is very impressive, I think it also fair to point out that growth rate is not the only predictor of fitness in yeast (though it is certainly the easiest to measure in high-throughput). At the very least, I’d expect to see some acknowledgement of this point somewhere in the manuscript, and perhaps this could be included in the Supplementary Note that enumerates some of the study’s limitations.

We agree that factors such as lag time and yield are also fitness components and have clarified that the maximum growth rate is used only as a proxy for fitness (lines 114-116).

4. I would have valued more discussion of within-treatment variation. Clearly, the point here is to identify adaptive alleles/genes that are shared across treatments, but your data may shed light on the question of whether parallel evolution across replicates might be environment-specific. Related to this point – you define 2 general treatment classes in this study (chemicals vs. more “natural” stressors). It wasn’t obvious why these two classes were designated, and the degree to which they may have been analyzed separately. For example, the PDR genes; these are ABC transporters that are expected to evolve in response to chemical/drug stress. PDR3, PDR1, and PDR5 are all included in the list of “general adaptation genes”, but is this due to repeatedly showing up in the chemical stress treatments, or did they indeed show up across all treatments? Regarding the fitness data, it would also be interesting to see if the fitness changes were greater overall in one treatment class versus another. Perhaps this could be incorporated into Figure 1a, with some type of color coding to show those environments belonging to each class.

We appreciate this constructive comment. For three main reasons, we did not extensively compare the adaptive genes across classes/groups of environments. First, the classification of environments into two broad classes was somewhat arbitrary. This is especially true to Class II environments, because they are a heterogenous set of chemicals that were further divided into 10 groups based on the fitness profiling of approximately 5,900 yeast gene deletion strains. Even within the same group, the different environments could be quite different. Second, we found no

significant difference in the mean Dice's coefficient of similarity in putatively adaptive genes between different environments within a group and those between groups (Fig. S5c), suggesting that the adaptive genetic responses observed are not substantially different among groups of environments. Finally, the primary focus of the study is to elucidate general patterns in the genomic basis of adaptation rather than delving into the specifics of which genes are utilized in response to which environment. Notwithstanding, we followed the reviewer's suggestion to analyze whether there are differences in adaptive genes across classes/groups of environments. The results have been added to the main text (lines 275-285), as shown below.

“Finally, we compared the adaptive genes identified in the two large classes of environments used in our experimental evolution (Fig. S1). We observed a higher prevalence of PDR genes (PDR1, PDR3, and PDR5) in Class II environments, whereas signal transduction genes such as IRA2, CYR1, and RAS2 are enriched in Class I environments. Specifically, 34% (52) of Class II environments but only 3% (3) of Class I environments used PDR genes ($P = 1.1 \times 10^{-7}$, chi-squared test). Conversely, 66% of Class I environments but only 52% of Class II environments used signal transduction genes ($P = 0.065$). Further subdivision of Class I environments into five groups (Fig. S1b) revealed a propensity for transmembrane genes in environments with different nitrogen resources, notably GNP1. Specifically, 41% (9) of environments with different nitrogen resources utilized GNP1, compared to only 15% in all other environments studied ($P = 0.0052$).”

We have followed the reviewer's suggestion to add colors to Fig. 1ab to distinguish between the two classes of environments (see below for the colored version, which is also presented as Fig. S2). However, we did not find any apparent patterns. For instance, the extent of adaptation (f_i/F_i) is not significantly different between the two classes of environments ($P = 0.47$, two-tailed Wilcoxon rank-sum test).

Minor comments

Line 83: “this enrichment may be attributable to a higher power of the approach in detecting diversifying selection than directional selection instead of an oversized contribution of these genes to adaptation”. This sentiment is not clear to me as written, can you rephrase? I don’t think of directional selection as difficult to detect, just rarely detected.

Under diversifying selection, orthologous genes from multiple species often show more than two states at a codon. By contrast, orthologous genes from multiple species typically show at most two states at a codon when under directional selection. When d_N/d_S is used to detect positive selection, it is usually those codons with more than two states that provide the strongest signal of positive selection. Consequently, diversifying selection is easier than directional selection to be detected in comparative genomics. We have added the above explanation to the main text as follows (lines 71-77).

“For instance, estimating the nonsynonymous to synonymous nucleotide substitution rate ratio (ω) in comparative genomics tends to detect positive selection in genes for immunity or reproduction⁷, but this enrichment may be attributable to a higher power of the approach in detecting diversifying selection than directional selection instead of an oversized contribution of these genes to adaptation, because diversifying selection tends to result in multiple nonsynonymous substitutions within a codon, facilitating the detection of positive selection.”

Throughout the manuscript “adaptations” is often referred to as plural when singular works better (for example, in the title).

Changed as suggested.

First sentence in Discussion is too speculative. According to my understanding, this study does not show that there is an increasing beneficial mutation rate with stress level, there is just a literature precedent to support this point. The study does test the idea that the effect of a mutation increases with stress level so that part is OK.

We have now provided a reference to the point of “an increased beneficial mutation rate” so that it is clear that this point is based on a previous study (lines 344-346).

Lines 160-161: I understand what is being stated here, but the language could be clearer. I think this statement could be clarified by simply stating that the higher substitution rate suggests stronger positive selection on coding regions in the observed data.

We believe that our statement should not be simplified as the one suggested by the reviewer, because an increased rate of substitution in coding regions could also be due to a relaxation of purifying selection.

Line 343: “agnostic” works better than “agonistic”, typo?

We have changed it to “not dependent on”.

Lines 353: appear to*

Changed as suggested.

Line 425: the “Experimental evolution” section of the methods seems a bit cursory. For example, was there some liquid handling automation (I would think this is likely given the high throughput of the experiment)? If so this is not mentioned.

We have provided the additional details as suggested (lines 451-454)

“Every 24 hrs, after the culture had reached the stationary phase, we transferred 2 μ l of stationary culture (around 10^5 cells) into 500 μ l fresh culture medium using Integra VIAFLO 96-channel handheld electronic pipette.”

Line 435: “we performed experimental evolution of 84 additional replicates in each of four environments chosen from the 252 environments”....to assess the impact of # reps on Dice’s

coefficient of similarity. I am not finding much discussion of this part of the experiment in the main text. This is related to major concern #4 – you seem to have measured the degree of evolutionary parallelism across replicates at high depth (at least for 4 treatments), but do not focus on that outcome.

The reviewer asked us to examine the parallelism across the 96 replicates in the four environments studied. We have followed the suggestion to analyze the adaptive genes in the four environments. Indeed, we identified more adaptive genes than those found in environments with 12 replicates, attributable to the significantly enhanced detection power provided by more parallel lines. We have now added the following (lines 230-232).

“As expected, with an increased detection power bestowed by 96 replicates, we observed more adaptive genes (8 to 15 per environment, with a median of 11.5 across the four environments).”

REVIEWERS' COMMENTS

Reviewer #1 (Remarks to the Author):

The authors have done a great job of addressing all of my concerns. I have checked the links provided for data availability and they in good working order. I have no further questions or comments, congratulations on a nice piece of work!

Reviewer #1 (Remarks on code availability):

I have confirmed that the code is available, but not run it myself.

Reviewer #2 (Remarks to the Author):

The authors have done meaningful changes to the manuscript, addressing or alleviating most of my concerns. I would have welcomed a more extensive analysis of to what extent the key patterns holds true across selection pressure categories, "nitrogen sources", "carbon sources", "small molecule" etc. for which the biology is rather distinct and well known. But I do not really doubt that the highlighted key patterns are broadly shared across environments and that authors' findings, interpreted with the cautions given in the discussion and the supplementary note, are mostly robust. Thus, I recommend Nature Communication to accept the paper.

Reviewer #3 (Remarks to the Author):

I appreciate the authors' consideration of my feedback, and I find the revised manuscript (which I already quite liked) improved in clarity and attention to detail. I support publication of the revised manuscript, and I expect it will be well-cited in the years to come.

Reviewer #3 (Remarks on code availability):

I did not install/run the code myself, but it appears to be well-annotated and modular.